# A motor neuron disease-associated mutation produces non-glycosylated Seipin that induces ER stress and apoptosis by inactivating SERCA2b

Shunsuke Saito, Tokiro Ishikawa, Satoshi Ninagawa[†], Tetsuya Okada, Kazutoshi Mori*

Department of Biophysics, Graduate School of Science, Kyoto University, Kyoto, Japan

**Abstract** A causal relationship between endoplasmic reticulum (ER) stress and the development of neurodegenerative diseases remains controversial. Here, we focused on Seipinopathy, a dominant motor neuron disease, based on the finding that its causal gene product, Seipin, is a protein that spans the ER membrane twice. Gain-of-function mutations of Seipin produce non-glycosylated Seipin (ngSeipin), which was previously shown to induce ER stress and apoptosis at both cell and mouse levels albeit with no clarified mechanism. We found that aggregation-prone ngSeipin dominantly inactivated SERCA2b, the major calcium pump in the ER, and decreased the calcium concentration in the ER, leading to ER stress and apoptosis in human colorectal carcinoma-derived cells (HCT116). This inactivation required oligomerization of ngSeipin and direct interaction of the C-terminus of ngSeipin with SERCA2b, and was observed in Seipin-deficient neuroblastoma (SH-SY5Y) cells expressing ngSeipin at an endogenous protein level. Our results thus provide a new direction to the controversy noted above.

*For correspondence:
mori@upr.biophys.kyoto-u.ac.jp

Present address: †Biosignal Research Center, Kobe University, Kobe, Japan

Competing interest: The authors declare that no competing interests exist.

## Editor's evaluation

Seipin is a multifunctional endoplasmic reticulum-localized protein associated with seemingly unrelated human diseases. Here, the authors establish a correlation between the expression of a particular mutant form of Seipin associated in humans with motor neuron disease and altered intracellular calcium dynamics and allied proteotoxic stress. The article is noted for the clues it provides into how these cellular defects arise and for offering a plausible, but yet unproven hypothesis for the cellular pathology that may account for the human disease phenotype.

## Introduction

The *BSCL1* and *BSCL2* genes have been identified as the causal genes of congenital generalized lipodystrophy (CGL) or Berardinelli–Seip congenital lipodystrophy syndrome (BSCL), a rare autosomal-recessive disease characterized by insufficiency of adipose tissue from birth or early infancy and by severe insulin resistance. The *BSCL1* gene encodes 1-acylglycerol-3-phosphate O-acyltransferase 2 (AGPAT2), which is present in the membrane of the endoplasmic reticulum (ER) and involved in phospholipid biosynthesis (*Agarwal et al., 2002*; *Garg et al., 1999*), whereas the *BSCL2* gene encodes Seipin, a protein which spans the ER membrane twice and whose function was unknown at that time (*Magré et al., 2001*). Loss-of-function mutations of the *BSCL2* gene appear to produce more severe symptoms than those of the *BSCL1* gene (*Van Maldergem et al., 2002*). Since the discovery that

Seipin is involved in lipid droplet morphology in yeast (*Szymanski et al., 2007*), the role of Seipin in the biogenesis of lipid droplets has gained extensive attention (*Bi et al., 2014*; *Cui et al., 2011*; *Sim et al., 2013*; *Sui et al., 2018*; *Tian et al., 2011*; *Wang et al., 2014*; *Wang et al., 2016*; *Yan et al., 2018*).

To our interest, two missense mutations of the *BSCL2* gene, namely, N152S and S154L of Seipin, were found to dominantly cause distal hereditary motor neuropathy (dHMN) or distal muscular atrophy, which is characterized almost exclusively by the degeneration of motor nerve fibers, predominantly in the distal part of limbs (*Windpassinger et al., 2004*). Because $N^{152}$, $V^{153}$, and $S^{154}$ of Seipin match the triplet code (Asn-X-Ser/Thr; X: any amino acid except Pro) for *N*-glycosylation, neither N152S nor S154L Seipin are glycosylated, leading to the proposal that the production of these aggregation-prone mutants results in neurodegeneration (*Windpassinger et al., 2004*).

Seipin was first identified as a protein of 398 aa (*Magré et al., 2001*), and later found (*Lundin et al., 2006*) to have two splice variants, a short form of 398 aa and long form of 462 aa (see *Figure 1A*), which are translated from three Seipin mRNA isoforms of 1.6 kb, 1.8 kb, and 2.2 kb. Both forms are translatable from 1.8 kb and 2.2 kb mRNA, but the long form is more abundantly produced than the short form. In contrast, only the short form is translated from 1.6 kb mRNA (*Lundin et al., 2006*). Because 1.8 kb mRNA is predominantly expressed in human brain (*Magré et al., 2001*), it is considered that human brain expresses mainly the long form (*Cartwright and Goodman, 2012*).

The ER, where Seipin is located, is well known to be equipped with a quality control system for proteins. Productive folding of newly synthesized secretory and transmembrane proteins is assisted by ER-localized molecular chaperones and folding enzymes (ER chaperones hereafter). In contrast, proteins unable to gain their correct three-dimensional structures are dealt with by ER-associated degradation (ERAD), in which unfolded or misfolded proteins are recognized, delivered to the transmembrane complex termed the retrotranslocon, and retrotranslocated to the cytosol for ubiquitin-dependent proteasomal degradation.

Under a variety of physiological and pathological conditions, however, this quality control system misfunctions, resulting in the accumulation of unfolded or misfolded proteins in the ER. This ER stress is quite detrimental to the cell and may eventually cause cell death. In response, ER stress is immediately and adequately counteracted by a cellular homeostatic mechanism termed the unfolded protein response (UPR). In vertebrates, the UPR is triggered by three types of ubiquitously expressed ER stress sensors and transducers – PERK, ATF6, and IRE1 – which in turn lead to general translational attenuation to decrease the burden on the ER; transcriptional upregulation of ER chaperones to increase productive folding capacity; and transcriptional upregulation of ERAD components to increase degradation capacity.

Daisuke Ito and colleagues showed for the first time that expression of non-glycosylated mutant Seipin by transfection in HeLa cells evokes ER stress, as evidenced by induction of the two major ER chaperones BiP and GRP94, the ERAD component Herp, and CHOP. HeLa cells expressing non-glycosylated mutant Seipin by transfection are subject to more extensive apoptosis (18%) than those expressing wild-type (WT) Seipin (6%) (*Ito and Suzuki, 2007*). Based on these findings, they proposed the designation of mutant Seipin-linked dominant motor neuron disease as Seipinopathy, which represents a novel ER stress-associated disease (*Ito and Suzuki, 2009*).

They further constructed a transgenic mice overexpressing human non-glycosylated mutant Seipin under the control of the neuron-specific murine Thy-1 promoter. They found that the levels of the ER stress marker proteins BiP and PDI are elevated in brain in these transgenic mice, reproducing the symptomatic and pathological phenotypes observed in human patients with Seipinopathy (*Yagi et al., 2011*).

Here, we focused on the remaining and highly critical question of how non-glycosylated mutant Seipin evokes ER stress.

## Results

### Construction of Seipinopathy-causal mutant Seipin

We found that human HCT116 diploid cells derived from colorectal carcinoma (*Roschke et al., 2002*), which we use exclusively for gene knockout analysis, expressed only the short form of Seipin, designated Seipin$^S$, whereas human neuroblastoma-derived SH-SY5Y cells with trisomy 7 (*Yusuf et al.,*

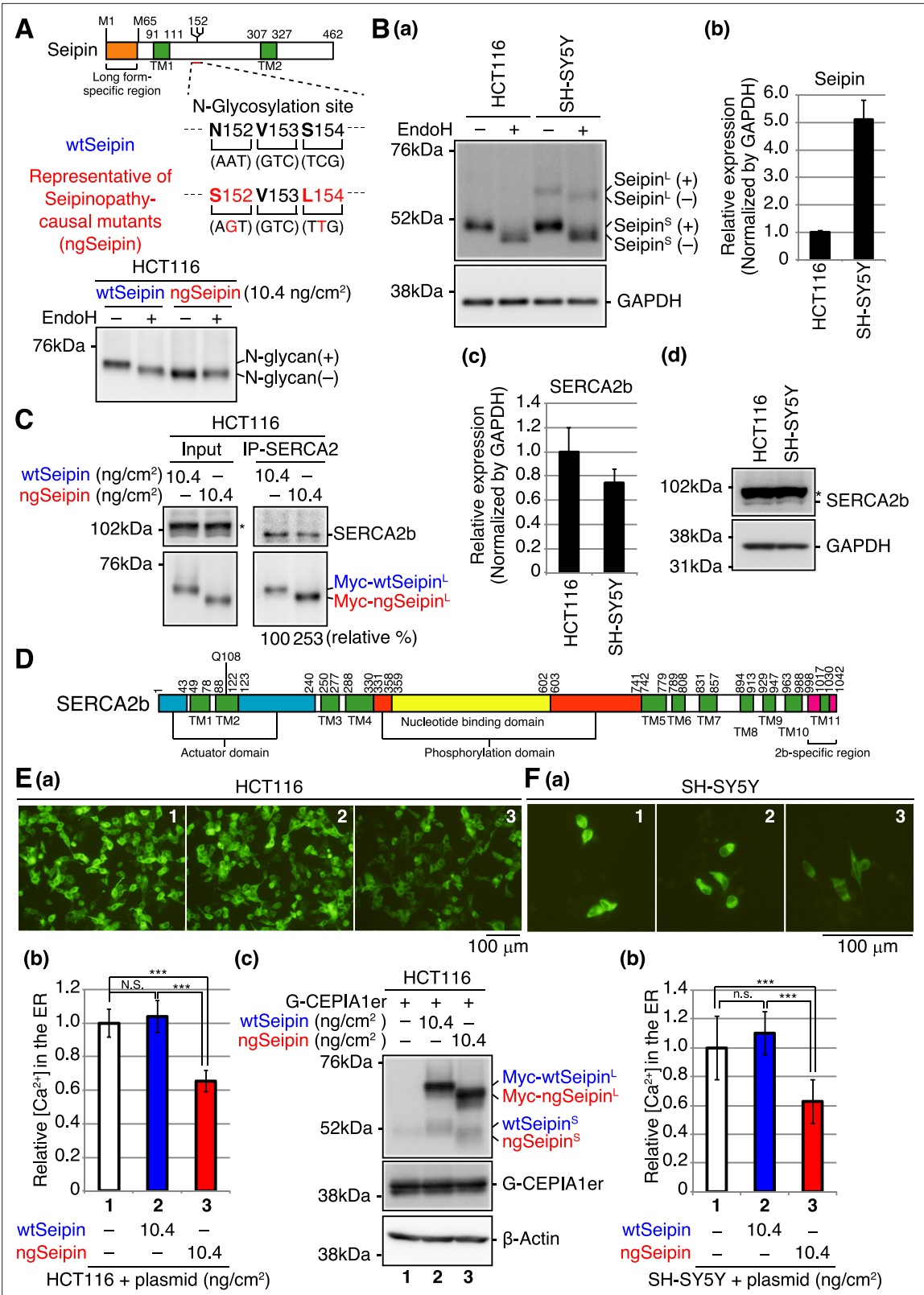

**Figure 1.** Effect of non-glycosylated Seipin (ngSeipin) expression on calcium concentration in the endoplasmic reticulum (ER) of HCT116 and SH-SY5Y cells. (**A**) Structure of human Seipin is schematically shown. The orange and green boxes denote the long form-specific region and transmembrane (TM1 and TM2) domains, respectively. The amino acid sequences of 152–154 and corresponding nucleotide sequences of wild-type Seipin (wtSeipin) and representative of Seipinopathy-causal mutants (ngSeipin) are shown below. Cell lysates were prepared from HCT116 cells transfected with plasmid

*Figure 1 continued on next page*

Figure 1 continued

(10.4 ng/cm²) to express Myc-tagged wtSeipin or ngSeipin (both long forms), treated with (+) or without (-) EndoH, and analyzed by immunoblotting using anti-Myc antibody. (B) (a) Cell lysates were prepared from HCT116 and SH-SY5Y cells, treated with (+) or without (-) EndoH, and analyzed by immunoblotting using anti-Seipin and anti-GAPDH antibodies. (+) and (-) denote glycosylated and de-glycosylated protein, respectively. (b) Quantitative RT-PCR was conducted to determine the level of endogenous Seipin mRNA (a region shared by Seipin$^L$ and Seipin$^S$ was amplified) relative to that of GAPDH mRNA in HCT116 and SH-SY5Y cells (n = 3). (c) Quantitative RT-PCR was conducted to determine the level of SERCA2b mRNA relative to that of GAPDH mRNA in HCT116 and SH-SY5Y cells (n = 3). (d) Cell lysates were prepared from HCT116 and SH-SY5Y cells and analyzed by immunoblotting using anti-SERCA2 and anti-GAPDH antibodies. The asterisk denotes a nonspecific band. (C) Cell lysates were prepared from HCT116 cells transfected with plasmid (10.4 ng/cm²) to express Myc-tagged wtSeipin or ngSeipin, and subjected to immunoprecipitation using anti-SERCA2 antibody. Aliquots of cell lysates (Input) and immunoprecipitates (IP-SERCA2) were analyzed by immunoblotting using anti-SERCA2 and anti-Myc antibodies. The asterisk denotes a nonspecific band. (D) Structure of human SERCA2b is schematically shown. Green, blue, orange, yellow, and red boxes denote transmembrane (TM1-TM11) domains, actuator domain, phosphorylation domain, nucleotide-binding domain, and SERCA2b-specific region, respectively. (E) HCT116 cells were transfected with plasmid (104 ng/cm²) to express G-CEPIA1er together with or without plasmid (10.4 ng/cm²) to express Myc-tagged wtSeipin or ngSeipin. (a) Fluorescent microscopic analysis of transfected cells was conducted. Scale bar: 100 μm. (b) Fluorescence intensities of 16–17 pictures obtained from three independent experiments (5–7 pictures each) were quantified and are expressed relative to that in cells transfected with plasmid to express G-CEPIA1er alone with error bars (standard deviation). (c) Cell lysates were prepared and analyzed by immunoblotting using anti-Seipin, anti-GFP, and anti-β-actin antibodies. (F) (a) Fluorescent microscopic analysis of SH-SY5Y cells transfected with plasmid (104 ng/cm²) to express G-CEPIA1er together with or without plasmid (10.4 ng/cm²) to express Myc-tagged wtSeipin or ngSeipin was conducted. Scale bar: 100 μm. (b) Fluorescence intensities of 15–17 pictures obtained from three independent experiments (5–7 pictures each) were quantified and are expressed as in (E). See also *Figure 1—source data 1*.

The online version of this article includes the following source data and figure supplement(s) for figure 1:

**Source data 1.** Raw data related to *Figure 1A, B, C and E*.

**Figure supplement 1.** Expression levels of Seipin mRNA and SERCA2 mRNA in different human tissues.

**Figure supplement 2.** Comparison of expression levels of SERCA1/2/3 mRNA and SERCA2a/b/c mRNA in HCT116 and SH-SY5Y cells as well as the effect of non-glycosylated Seipin (ngSeipin) expression on RyR and IP3R.

---

*2013*) expressed both Seipin$^S$ and the long form of Seipin, designated Seipin$^L$, and further showed that Seipin$^S$ and Seipin$^L$ were both sensitive to digestion with endoglycosidase H (Endo H) (*Figure 1B(a)*; note that the same amounts of total proteins [10 μg] in cell lysates were analyzed). When normalized by the level of GAPDH, which was quite similarly detected by immunoblotting in cell lysates prepared from the same number (1.2 × 10⁴ cells) of HCT116 and SH-SY5Y cells (data not shown), quantitative RT-PCR showed that SH-SY5Y cells expressed Seipin mRNA five times more abundantly than HCT116 cells (*Figure 1B(b)*). Of note, searching the RNA sequencing database 'Expression Atlas: Gene expression across species and biological conditions' (https://www.ebi.ac.uk/gxa/home) showed that, in humans, Seipin mRNA is highly expressed in the nervous system compared with other tissues except for testis, whereas SERCA2 mRNA is relatively ubiquitously expressed (*Figure 1—figure supplement 1*; *GTEx Consortium, 2015*; *Papatheodorou et al., 2020*).

To produce the representative of Seipinopathy-causal mutants, we simultaneously mutated Asn$^{152}$ and Ser$^{154}$ of Seipin$^L$ to Ser and Leu, respectively (*Figure 1A*, middle). When expressed in HCT116 cells by transfection, N-terminally Myc-tagged WT Seipin$^L$ was sensitive to digestion with Endo H but N-terminally Myc-tagged mutant (N152S/S154L) Seipin$^L$ was not, as expected (*Figure 1A*, bottom); WT Seipin$^L$ and the non-glycosylated mutant Seipin$^L$ are hereafter designated wtSeipin$^L$ and ngSeipin$^L$, respectively. It should be noted that, because we carried out transfection in cell culture systems of various sizes, we express a transfection index as the amount of plasmid (ng) divided by the bottom area (cm²) of the well/dish, that is, 2.0 cm² for a 24-well plate, 9.6 cm² for a 6-well plate, and 11.8 cm² for a 3.5 cm dish, for easier comparison of results obtained from different experiments; accordingly, when 100 and 123 ng plasmid was transfected into cells in 6-well plates and 3.5 cm dishes, respectively, the transfection index was 10.4 ng/cm².

## Effect of ngSeipin expression on calcium concentration in the ER

We focused on SERCA2, the major calcium pump in the ER incorporating cytosolic calcium ion into the ER, based on the previous observation that Seipin physically associates with SERCA in fly as well as with SERCA2 in HEK293 cells (*Bi et al., 2014*). It should be noted that the expression level of SERCA2 dominated that of SERCA1 and SERCA3 in both HCT116 and SH-SY5Y cells (*Figure 1—figure supplement 2A(a, b)*). It is also known that three splice variants exist for SERCA2 (*Gélébart et al., 2003*) and that SERCA2b is ubiquitously expressed, whereas SERCA2a and SERCA2c are expressed mainly

in myocardium and skeletal muscle, in which the expression level of SERCA2b is low (*Dally et al., 2006*). Indeed, the expression level of SERCA2b dominated that of SERCA2a and SERCA2c in both HCT116 and SH-SY5Y cells (*Figure 1—figure supplement 2B(a, b)*). Of note, quantitative RT-PCR showed that the level of SERCA2b mRNA in SH-SY5Y cells was comparable with that in HCT116 cells (*Figure 1B(c)*) and immunoblotting showed that the level of SERCA2b protein in SH-SY5Y cells was also comparable with that in HCT116 cells (*Figure 1B(d)*). Interestingly, immunoprecipitation from HCT116 cells expressing wtSeipin$^L$ or ngSeipin$^L$ by transfection showed that ngSeipin$^L$ bound to SERCA2b more extensively than wtSeipin$^L$ (*Figure 1C*). The structure of SERCA2b is schematically shown in *Figure 1D*.

To monitor calcium concentration ([Ca$^{2+}$] hereafter) in HCT116 cells, we employed the fluorescent reporters G-CEPIA1er and GCaMP6f, whose fluorescence reflects [Ca$^{2+}$] in the ER (*Suzuki et al., 2014*) and in cytosol (*Chen et al., 2013*), respectively. Expression of ngSeipin$^L$ in HCT116 cells by transfection (10.4 ng/cm$^2$) markedly decreased [Ca$^{2+}$] in the ER compared with that of wtSeipin$^L$ (*Figure 1E(a, b)*). This effect of ngSeipin$^L$ on [Ca$^{2+}$] in the ER was also observed in SH-SY5Y cells (*Figure 1F(a, b)*). We noted that small amounts (~20%) of wtSeipin$^S$ and ngSeipin$^S$ were produced from transfected plasmid (*Figure 1E(c)*). Because they were not detected with anti-Myc antibody (data not shown), it is likely that they were translated from the second methionine M$^{65}$ (see *Figure 1A*), given that the nucleotide sequences around M65 (ccgGccATGG) are more similar to the Kozak consensus sequence for translational initiation (gccRccATGG) than those around M1 (aggAagATGt).

## Specific and dominant inactivation of SERCA2b by ngSeipin

The ER contains two types of calcium channel, namely, ryanodine receptor (RyR) and IP3 receptor (IP3R), which release stored calcium to the cytosol upon various stimuli, for example, 4-chloro-m-cresol (4CmC) for RyR (*Zorzato et al., 1993*) and bradykinin for IP3R (*Cruzblanca et al., 1998*). Quantitative RT-PCR detected expression of mRNA encoding RyR1, IP3R1, IP3R2, and IP3R3 in HCT116 cells (*Figure 1—figure supplement 2C*). 4CmC, bradykinin, or both decreased [Ca$^{2+}$] in the ER in untransfected HCT116 cells and in HCT116 cells expressing wtSeipin$^L$ by transfection (10.4 ng/cm$^2$), as expected (*Figure 1—figure supplement 2D(a–d)*, white and blue bars). The lowered [Ca$^{2+}$] in the ER of HCT116 cells expressing ngSeipin$^L$ by transfection (10.4 ng/cm$^2$) compared with those expressing wtSeipin$^L$ (*Figure 1—figure supplement 2D(d)*; compare bars 5, 11, and 17 with bars 3, 9, and 15) was further decreased upon treatment with 4CmC, bradykinin, or both (*Figure 1—figure supplement 2D(d)*; compare bar 6 with bar 5, bar 12 with bar 11, bar 18 with bar 17), suggesting that RyR and IP3R are still active in HCT116 cells expressing ngSeipin$^L$.

Thapsigargin treatment rapidly increases [Ca$^{2+}$] in cytosol by inhibiting SERCA1/2/3 without affecting RyR or IP3R (*Lytton et al., 1991*). The total amount of calcium released from the ER to cytosol upon thapsigargin treatment, which was monitored using GCaMP6f, was markedly decreased in HCT116 cells expressing ngSeipin$^L$ by transfection (10.4 ng/cm$^2$) compared with those expressing wtSeipin$^L$ (*Figure 2A*), reflecting lowered [Ca$^{2+}$] in the ER (*Figure 1E*), which was monitored using G-CEPIA1er. The treatment of HCT116 cells or HCT116 cells expressing wtSeipin$^L$ by transfection (10.4 ng/cm$^2$) with CDN1163, a SERCA2 activator (*Cornea et al., 2013*; *Gruber et al., 2014*), increased [Ca$^{2+}$] in the ER, as expected, whereas the treatment of HCT116 cells expressing ngSeipin$^L$ by transfection (10.4 ng/cm$^2$) with CDN1163 did not do (*Figure 2—figure supplement 1A*). These results suggest that ngSeipin$^L$ selectively inactivates SERCA2b.

Because Seipinopathy is an autosomal-dominant disease, we next examined the effect of co-expression of wtSeipin$^L$ and ngSeipin$^L$ on [Ca$^{2+}$] in the ER. Expression of ngSeipin$^L$ by transfection at 5.20 and 10.4 ng/cm$^2$ decreased [Ca$^{2+}$] in the ER in a dose-dependent manner (*Figure 2B(a, b)*; compare bars 3 and 8 with bar 1), whereas expression of wtSeipin$^L$ by transfection at 5.20 and 10.4 ng/cm$^2$ did not do (*Figure 2B(a, b)*; compare bars 2 and 4 with bar 1). Co-expression of wtSeipin$^L$ in a decreasing manner by transfection at 7.80, 5.20, and 2.60 ng/cm$^2$ and of ngSeipin$^L$ in an increasing manner by transfection at 2.60, 5.20, and 7.80 ng/cm$^2$ (transfection at a total of 10.4 ng/cm$^2$) decreased [Ca$^{2+}$] in the ER (*Figure 2B(a, b)*; compare bars 5, 6, and 7 with bar 1). Furthermore, CDN1163 treatment did not increase [Ca$^{2+}$] in the ER significantly in HCT116 cells co-expressing wtSeipin$^L$ and ngSeipin$^L$ by transfection (5.20 ng/cm$^2$ each), in contrast to the case of untransfected HCT116 cells (*Figure 2—figure supplement 1B*). Thus, ngSeipin$^L$ dominantly inactivates SERCA2b and thereby dominantly decreases [Ca$^{2+}$] in the ER.

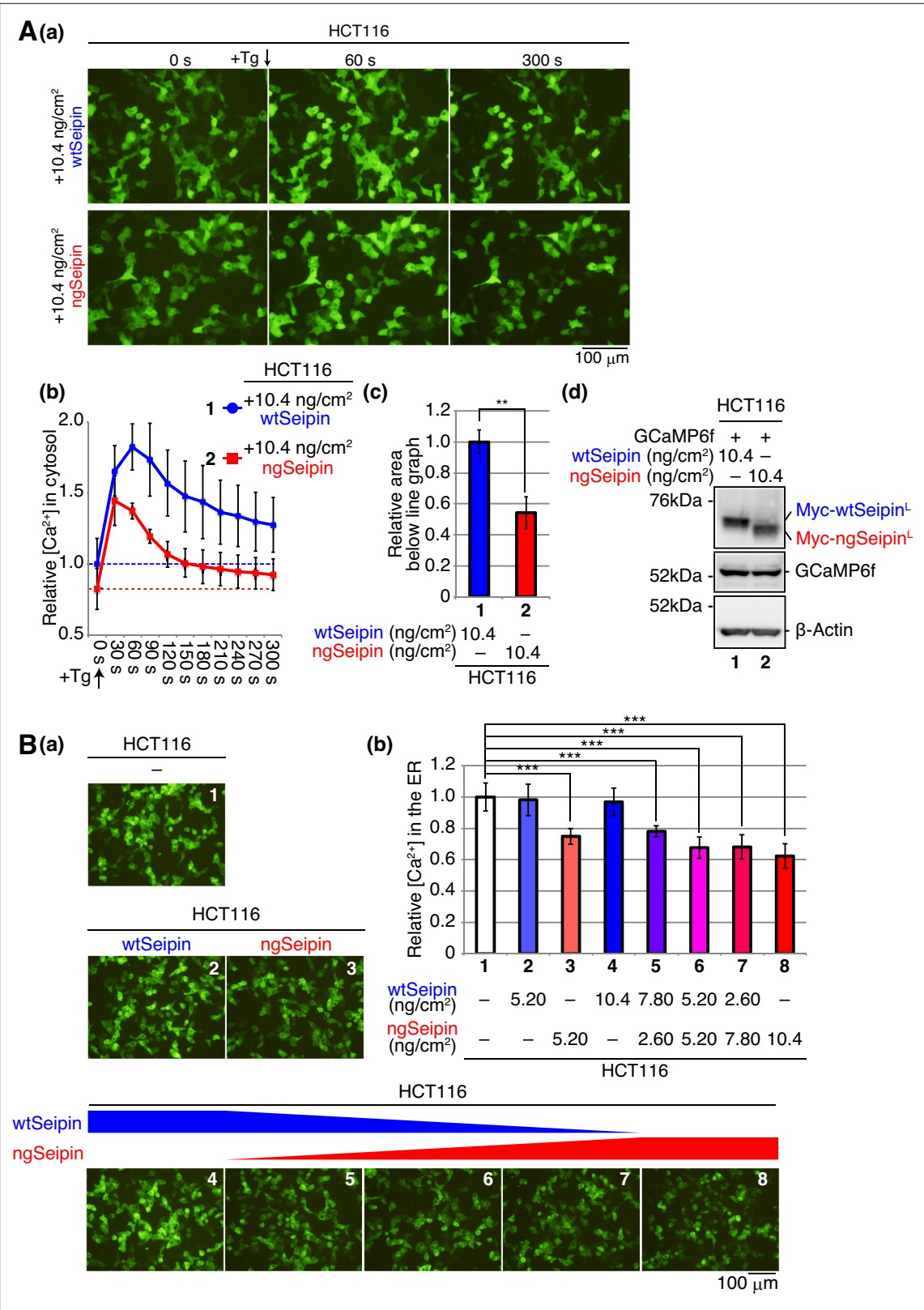

**Figure 2.** Effect of non-glycosylated Seipin (ngSeipin) expression on calcium concentration in the cytosol and effect of co-expression of ngSeipin and wild-type Seipin (wtSeipin) on calcium concentration in the endoplasmic reticulum (ER) in HCT116 cells. (**A**) HCT116 cells were transfected with plasmid (104 ng/cm²) to express GCaMP6f together with plasmid (10.4 ng/cm²) to express Myc-tagged wtSeipin or ngSeipin. (**a**) Fluorescent microscopic analysis was conducted before (0 s) and every 30 s after treatment with 1 μM thapsigargin (Tg). Only pictures of 0 s, 60 s, and 300 s are shown. Scale bar: 100 μm.

*Figure 2 continued on next page*

*Figure 2 continued*

(**b**) Fluorescence intensities were quantified at each time point and are shown as a line graph with the fluorescence intensity in HCT116 cells expressing wtSeipin before Tg treatment set as 1 (n = 3). (**c**) The area below the red line graph until the broken red line (fluorescence intensity at 0 s,+ngSeipin) in (**b**) was calculated and is shown relative to that below the blue line graph until the broken blue line (fluorescence intensity at 0 s,+wtSeipin). (**d**) Cell lysates were prepared and analyzed by immunoblotting using anti-Myc, anti-GFP, and anti-β-actin antibodies. (**B**) HCT116 cells were transfected with plasmid (104 ng/cm$^2$) to express G-CEPIA1er together with or without the indicated amounts of plasmid to express Myc-tagged wtSeipin or ngSeipin. (**a**) Fluorescent microscopic analysis was conducted. Scale bar: 100 μm. (**b**) Fluorescence intensities were quantified and are expressed as in *Figure 1E(b)* (n = 3). See also *Figure 2—source data 1*.

The online version of this article includes the following source data and figure supplement(s) for figure 2:

**Source data 1.** Raw data related to *Figure 2A*.

**Figure supplement 1.** Effect of CDN1163 on calcium concentration in the endoplasmic reticulum (ER) as well as construction of Seipin-KO HCT116 cells.

**Figure supplement 1—source data 1.** Raw data related to *Figure 2—figure supplement 1D, E*.

## Construction and characterization of Seipin-knockout cells

To examine the effect of endogenous Seipin on SERCA2b, we knocked out (KO) the *Seipin* gene in HCT116 cells using CRISPR/Cas9-mediated cleavage of the *Seipin* locus at two sites (*Figure 2—figure supplement 1C*). The deletion of almost the entire *Seipin* gene was confirmed by genomic PCR (*Figure 2—figure supplement 1D*), and the absence of *Seipin* mRNA and Seipin protein was confirmed by RT-PCR (*Figure 2—figure supplement 1E*) and immunoblotting (*Figure 3A(c)*; compare lane 2 with lane 1), respectively.

[Ca$^{2+}$] in the ER was decreased in Seipin-KO cells by 20–30% compared with WT cells (*Figure 3A(a, b)*; compare bar 2 with bar 1), consistent with an ~30% decrease in SERCA activity in Seipin mutant fly compared with WT fly and with Bi et al.'s proposal that Seipin aids the maintenance of calcium homeostasis in the ER by binding to SERCA (*Bi et al., 2014*). Expression of both wtSeipin$^L$ and ngSeipin$^L$ in Seipin-KO cells at a low level by transfection at 0.52 ng/cm$^2$, which was comparable with the level of endogenous Seipin$^S$ (*Figure 3A(c)*; compare lanes 3 and 4 with lane 1), restored [Ca$^{2+}$] in the ER (*Figure 3A(a, b)*; compare bars 3 and 4 with bars 1 and 2).

In contrast, a higher expression of ngSeipin$^L$ in Seipin-KO cells by transfection at ≥2.60 ng/cm$^2$ decreased [Ca$^{2+}$] in the ER in dose-dependent manner more robustly than that of wtSeipin$^L$ (*Figure 3A(a, b)*; compare bars 6, 8, and 10 with bars 5, 7, and 9). Higher expression of ngSeipin$^L$ by transfection at ≥2.60 ng/cm$^2$ induced ER stress more extensively than that of wtSeipin$^L$ in Seipin-KO cells, as evidenced by increased levels of BiP mRNA (a target of the ATF6 pathway), XBP1(S) mRNA (a target of the IRE1 pathway), and CHOP mRNA (a target of the PERK pathway) (*Figure 3B*; compare bars 6, 8, and 10 with bars 5, 7, and 9). Higher expression of ngSeipin$^L$ by transfection at ≥2.60 ng/cm$^2$ induced apoptosis more extensively than that of wtSeipin$^L$ in Seipin-KO cells, as shown by increased detection of cleaved Caspase-3 by immunofluorescence (*Figure 3C(a, b)*; compare bars 6, 8, and 10 with bars 5, 7, and 9). Of note, clear-cut difference in the effect on [Ca$^{2+}$] in the ER and apoptosis between transfection of ngSeipin at 0.52 ng/cm$^2$ (not significant) and ≥2.60 ng/cm$^2$ (significant) (*Figure 3A(a, b) and C(a, b)*) is best reflected in the induction of CHOP (*Figure 3B*), suggesting the importance of the PERK pathway in this ER stress-induced apoptosis. These results suggest that ngSeipin$^L$ expression-mediated decrease in [Ca$^{2+}$] in the ER is a key factor in the development of Seipinopathy.

## Effect of oligomerization of ngSeipin on inactivation of SERCA2b

To elucidate the mechanism by which ngSeipin$^L$ inactivates SERCA2b, we examined the effect of oligomerization of Seipin because human Seipin exists as a wheel-like undecamer (*Figure 4A*; *Yan et al., 2018*). Interestingly, fly Seipin consisting of 370 aa is not glycosylated and exists as a wheel-shaped dodecamer (*Sui et al., 2018*). Because *N*-glycan wedges the interface of two protomers in the case of human Seipin (*Figure 4A*, right panel), we hypothesized that non-glycosylated human Seipin expressed at a higher level becomes unable to maintain the undecamer structure, leading to aggregation.

Structural analysis revealed that the luminal region of each Seipin monomer consists of 8 β-strands, termed the β-sandwich domain, and 3 α-helices, and that an ER membrane-anchored core ring is

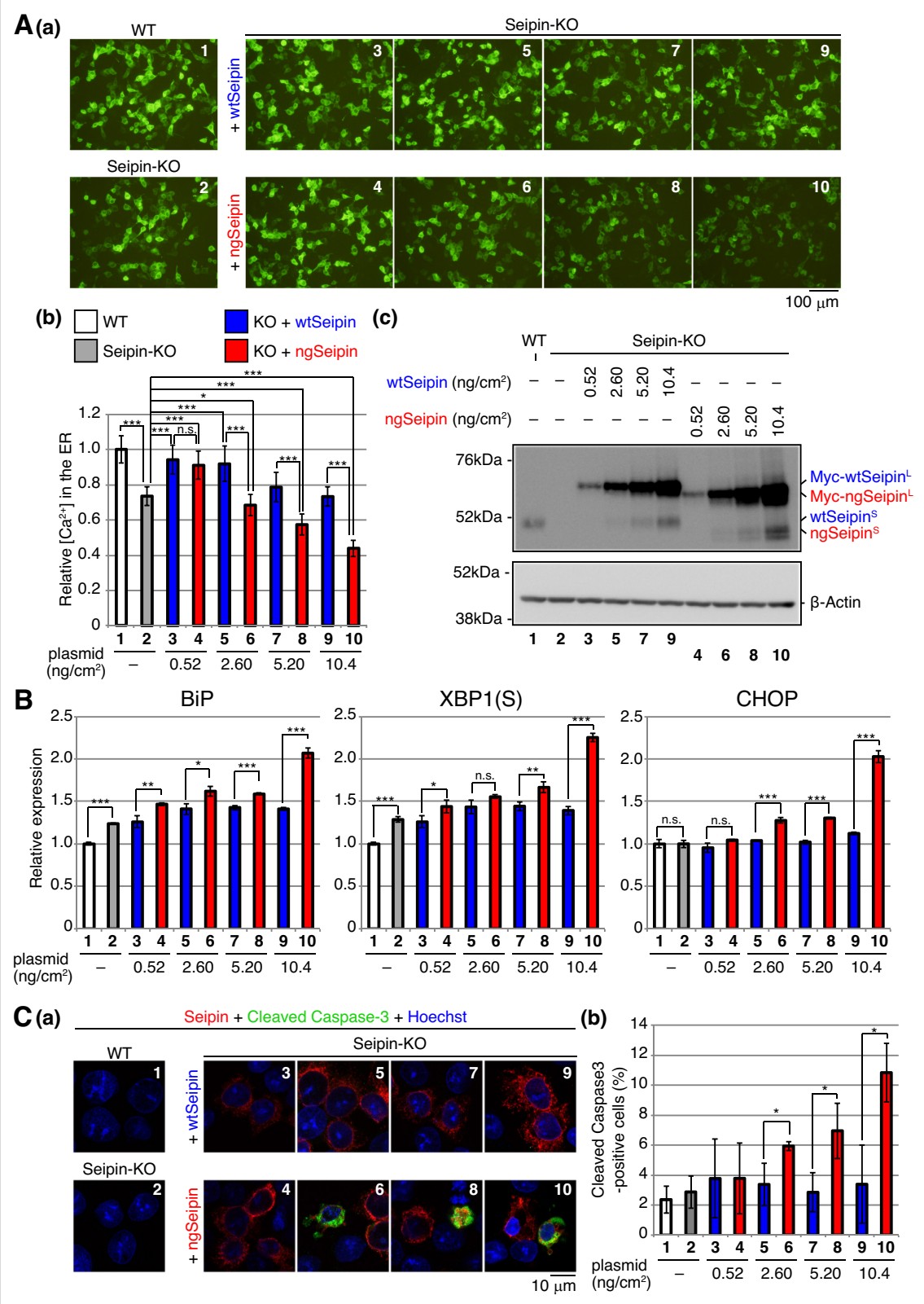

**Figure 3.** Effect of non-glycosylated Seipin (ngSeipin) expression on calcium concentration in the endoplasmic reticulum (ER), ER stress, and apoptosis in Seipin-KO HCT116 cells. (**A**) WT cells were transfected with plasmid (104 ng/cm²) to express G-CEPIA1er. Seipin-KO cells were transfected with plasmid (104 ng/cm²) to express G-CEPIA1er together with or without the indicated amounts of plasmid to express Myc-tagged wild-type Seipin (wtSeipin) or ngSeipin. (**a**) Fluorescent microscopic analysis of WT cells and Seipin-KO cells transfected as indicated was conducted. Scale bar:

*Figure 3 continued on next page*

*Figure 3 continued*

100 µm. (**b**) Fluorescence intensities were quantified and are expressed as in *Figure 1E(b)* (n = 3). (**c**) Cell lysates were prepared from the indicated cells and analyzed by immunoblotting using anti-Seipin and anti-β-actin antibodies. (**B**) Quantitative RT-PCR was conducted to determine the levels of endogenous BiP mRNA, spliced XBP1 [XBP1(S)] mRNA and CHOP mRNA relative to that of GAPDH mRNA in WT cells, Seipin-KO cells, and Seipin-KO cells transfected with the indicated amounts of plasmid to express Myc-tagged wtSeipin or ngSeipin (n = 3). The mean value of untransfected WT cells is set as 1. (**C**) (**a**) WT cells, Seipin-KO cells, and Seipin-KO cells transfected with the indicated amounts of plasmid to express Flag-tagged wtSeipin or ngSeipin were fixed 28 hr later, subjected to immunofluorescence using anti-Flag (red) and anti-cleaved Caspase-3 (green) antibodies, and analyzed by confocal microscopy. Scale bar: 10 µm. (**b**) The number of Flag-tagged Seipin and cleaved Caspase-3 double-positive cells was counted in 118–250 cells obtained from three independent experiments and is shown as a percentage. See also *Figure 3—source data 1*.

The online version of this article includes the following source data for figure 3:

**Source data 1.** Raw data related to *Figure 3A*.

formed when the 3 α-helices of each of 11 monomers are gathered via multiple hydrophobic interactions between one protomer and its neighboring protomer, including those between L226 and L220 and between L233 and V227. The core ring is surrounded by 11 β-sandwich domains that are tightly associated via hydrogen bonds between one protomer and its neighboring protomer, including those between S181 and Y215, between Q239 and S217, and between H131 and R209; and via hydrophobic interactions between one protomer and its neighboring protomer, including those between Y134 and M189 (*Figure 4A*, right panel; *Yan et al., 2018*). We therefore simultaneously mutated the six amino acids in wtSeipin$^L$ and ngSeipin$^L$ (H131R, Y134A, Y215A, L220D, L233D, and Q239A, *Figure 4B*), as previously reported to prevent oligomerization (*Yan et al., 2018*). Resulting Seipin$^L$(M6) and ngSeipin$^L$(M6) were still sensitive and insensitive, respectively, to digestion with EndoH, as expected (*Figure 4C*).

To confirm the aggregation propensity of ngSeipin, we employed a proximity ligation assay (PLA), in which a PCR-mediated signal is produced when the distance between two proteins is less than 40 nm (*Figure 4D*), and which was used to detect aggregates of α-synuclein, a causal protein of familial Parkinson's disease (*Roberts et al., 2015*). Results showed the production of a markedly strong signal when Myc-tagged ngSeipin$^L$ and Flag-tagged ngSeipin$^L$ were co-expressed by transfection each at $\geq$1.30 ng/cm$^2$ (total at $\geq$2.60 ng/cm$^2$) compared with co-expression of Myc-tagged wtSeipin$^L$ and Flag-tagged wtSeipin$^L$ (*Figure 4E(a, b)*; compare dot plot 4 with dot plot 3, dot plot 6 with dot plot 5, and dot plot 8 with dot plot 7), as we expected. Importantly, this signal was diminished when Myc-tagged ngSeipin$^L$(M6) and Flag-tagged ngSeipin$^L$(M6) were co-expressed by transfection each at 5.20 ng/cm$^2$ (total at 10.4 ng/cm$^2$) (*Figure 4E(a, b)*; compare dot plot 10 with dot plot 8), supporting our hypothesis of non-glycosylated Seipin oligomer (undecamer)-dependent aggregation when expressed at a higher level. Accordingly, although immunoprecipitation showed the association of ngSeipin$^L$(M6) with SERCA2b (*Figure 5A*), ngSeipin$^L$(M6) introduced into Seipin-KO cells by transfection even at 10.4 ng/cm$^2$ (highest transfection level in this report) did not decrease [Ca$^{2+}$] in the ER (*Figure 5B*), did not induce ER stress (*Figure 5C*), and did not induce apoptosis (*Figure 5D*), in marked contrast to ngSeipin$^L$. These results suggest that SERCA2b is incorporated into aggregates of oligomerized ngSeipin$^L$ to be inactivated. Of note, Flag-tagged ngSeipin$^L$ formed aggregates with Myc-tagged wtSeipin more extensively than Flag-tagged ngSeipin$^L$(M6) when analyzed by PLA (*Figure 4E(a, b)*; compare dot plot 11 with dot blot 13), indicative of its dominant nature.

## Requirement of both the luminal and C-terminal regions of ngSeipin for inactivation of SERCA2b

To determine which region(s) of ngSeipin$^L$ is required for inactivation of SERCA2b, we constructed a series of deletion mutants in wtSeipin$^L$ and ngSeipin$^L$, namely, ΔN lacking the cytosolic N-terminal region, ΔLD lacking the luminal region, and ΔC lacking the cytosolic C-terminal region (*Figure 6A*). We also constructed two swap mutants of wtSeipin$^L$ and ngSeipin$^L$, in which the first and second transmembrane (TM) domains of Seipin$^L$ were replaced by the fourth and first TM domains of glucose 6-phosphatase, respectively (*Figure 6A*), in reference to the previous swapping experiments (*Bi et al., 2014*) and with further consideration of the topology of these TM domains (*Figure 6B*). All constructs produced a band of the expected size in transfected WT cells (*Figure 6C* and *Figure 6—figure supplement 1A*, Input).

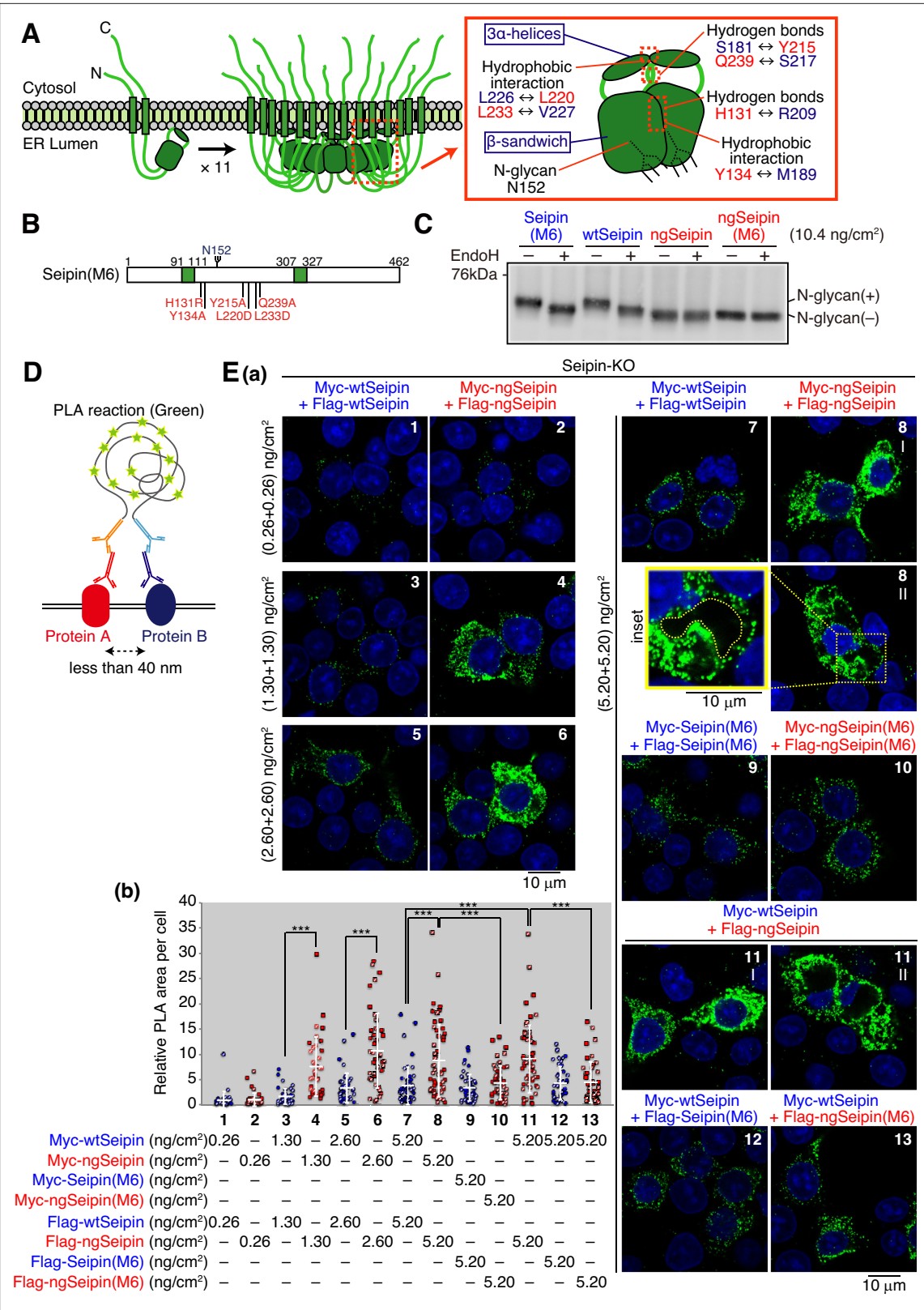

**Figure 4.** Luminal region-mediated aggregation of non-glycosylated Seipin (ngSeipin). (**A**) Structures of human Seipin monomer and undecamer are schematically shown on the left. Structure of one Seipin protomer and its neighboring protomer is schematically shown on the right with the positions of the β-sandwich domain, three α-helices, hydrophobic interactions, hydrogen bonds, and *N*-glycan. (**B**) Structure of Seipin(M6) is schematically shown, in which the six amino acids critical for oligomerization of Seipin are mutated as indicated and highlighted in red. (**C**) Cell lysates were prepared from

*Figure 4 continued*

HCT116 WT cells transfected with plasmid (10.4 ng/cm$^2$) to express Myc-tagged wild-type Seipin (wtSeipin), Seipin(M6), ngSeipin, or ngSeipin(M6), treated with (+) or without (-) EndoH, and analyzed by immunoblotting using anti-Myc antibody. (**D**) The principle of proximity ligation assay (PLA) is diagrammatically presented. PLA detects the proximal interaction (less than 40 nm) of two proteins in the cell using two different antibodies. (**E**) (**a**) PLA was conducted in Seipin-KO cells transfected with the indicated amounts of plasmid to simultaneously express Myc-tagged and Flag-tagged proteins with various combinations as indicated, and analyzed by confocal microscopy. Scale bar: 10 µm. (**b**) PLA signals were quantified in 35–57 cells obtained from two independent experiments (filled and striped dots denote the data of experiments 1 and 2, respectively) and are shown as signals (summation of PLA-positive area) per cell relative to those observed in cells transfected simultaneously with plasmid to express Myc-tagged wtSeipin (0.26 ng/cm$^2$) and plasmid to express Flag-tagged wtSeipin (0.26 ng/cm$^2$). See also *Figure 4—source data 1*.

The online version of this article includes the following source data for figure 4:

**Source data 1.** Raw data related to *Figure 4A*.

Immunoprecipitation using anti-SERCA2 antibody revealed that only Seipin$^L$(ΔC) and ngSeipin-$^L$(ΔC) were hardly co-immunoprecipitated with SERCA2b (*Figure 6C*, lanes 10 and 11, and *Figure 6—figure supplement 1A*, lanes 23 and 24). We thus constructed a mutant that expresses only the C-terminal region of Seipin$^L$ and found that this Seipin(Cterm) was efficiently co-immunoprecipitated with SERCA2b (*Figure 6C*, lane 12). These findings indicate that the cytosolic C-terminal region of Seipin$^L$ is necessary and sufficient for interaction with SERCA2b.

In the case of ΔN (*Figure 6D*; compare bar 6 with bar 5), TM1 (compare bar 8 with bar 7), and TM2 (compare bar 11 with bar 10) mutants, ngSeipin$^L$-based mutants decreased [Ca$^{2+}$] in the ER more extensively than wtSeipin$^L$-based mutants, similarly to the case of ngSeipin$^L$ versus wtSeipin$^L$ (compare bar 4 with bar 3), when expressed in Seipin-KO cells by transfection at 10.4 ng/cm$^2$. We later provide the reason for the marked decrease in [Ca$^{2+}$] in the ER in Seipin-KO cells expressing Seipin$^L$(ΔN), Seipin$^L$(TM1), and Seipin$^L$(TM2) by transfection to the level observed in Seipin-KO cells expressing ngSeipin$^L$ by transfection (*Figure 6D*, compare bar 4 with bars 5, 7, and 10). In contrast, ngSeipin$^L$(ΔC) lost the ability to decrease [Ca$^{2+}$] in the ER (*Figure 6D*; compare bar 13 with bar 12), indicating that direct interaction with SERCA2b is critical. Of note, Seipin$^L$(ΔLD) and Seipin(Cterm) did not decrease [Ca$^{2+}$] in the ER (*Figure 6D*, yellow bars 9 and 14), although they were co-immunoprecipitated with SERCA2b (*Figure 6—figure supplement 1A*, lane 20, and *Figure 6C*, lane 12). Given that the luminal region contains the six amino acids critical for oligomerization of Seipin (*Figure 4B*), both the luminal and C-terminal regions of ngSeipin$^L$ are required for inactivation of SERCA2b.

Interestingly, Myc-tagged ngSeipin$^L$(ΔC) and Flag-tagged ngSeipin$^L$(ΔC) co-expressed by transfection at a total of 10.4 ng/cm$^2$ (highest transfection level in this report) existed in closer proximity with each other than Myc-tagged Seipin$^L$(ΔC) and Flag-tagged Seipin$^L$(ΔC) co-expressed by transfection, as shown by PLA (*Figure 7A*, compare dot plot 4 with dot plot 3). Nonetheless, ngSeipin$^L$(ΔC) did not decrease [Ca$^{2+}$] in the ER (*Figure 6D*, compare bar 13 with bar 2), did not induce ER stress (*Figure 7B*), and did not induce apoptosis (*Figure 7C*). We concluded that the oligomerization-mediated aggregation propensity and direct interaction with SERCA2b are prerequisites for ngSeipin$^L$ to inactivate SERCA2b and thereby induce ER stress and subsequent apoptosis (*Figure 7D*).

## Effect of ngSeipin expression on morphology of the ER and localization of SERCA2b

Immunofluorescence showed that endogenous Seipin and SERCA2b were colocalized in the ER, which was stained by anti-KDEL antibody, although nuclear region was also stained with anti-SERCA2 antibody, consistent with the presence of SERCA2 in the nuclear envelope (*Abrenica and Gilchrist, 2000*; *Figure 7E*). Similarly, wtSeipin$^L$ expressed in Seipin-KO cells by transfection at 0.52–10.4 ng/cm$^2$ showed a typical ER pattern and was colocalized with endogenous SERCA2b (*Figure 8A(a)*, panels 1, 3, 5, and 7, transfected cells are surrounded by white broken lines).

In marked contrast, ngSeipin$^L$ expressed in Seipin-KO cells by transfection at 2.60–10.4 ng/cm$^2$ showed abnormal distribution in enlarged cells (*Figure 8A(a)*, panels 4, 6, and 8, transfected cells are surrounded by white broken lines), and accordingly endogenous SERCA2b also showed abnormal distribution. Representative cells expressing ngSeipin$^L$ by transfection at 10.4 ng/cm$^2$ contained strongly double-positive regions (*Figure 8A(b)*, indicated by white arrows), SERCA2b-positive-only regions surrounding the nucleus (*Figure 8A(b)*, indicated by asterisks), and multiple double-negative regions (*Figure 8A(b)*, surrounded by yellow broken lines). The percentage of cells with

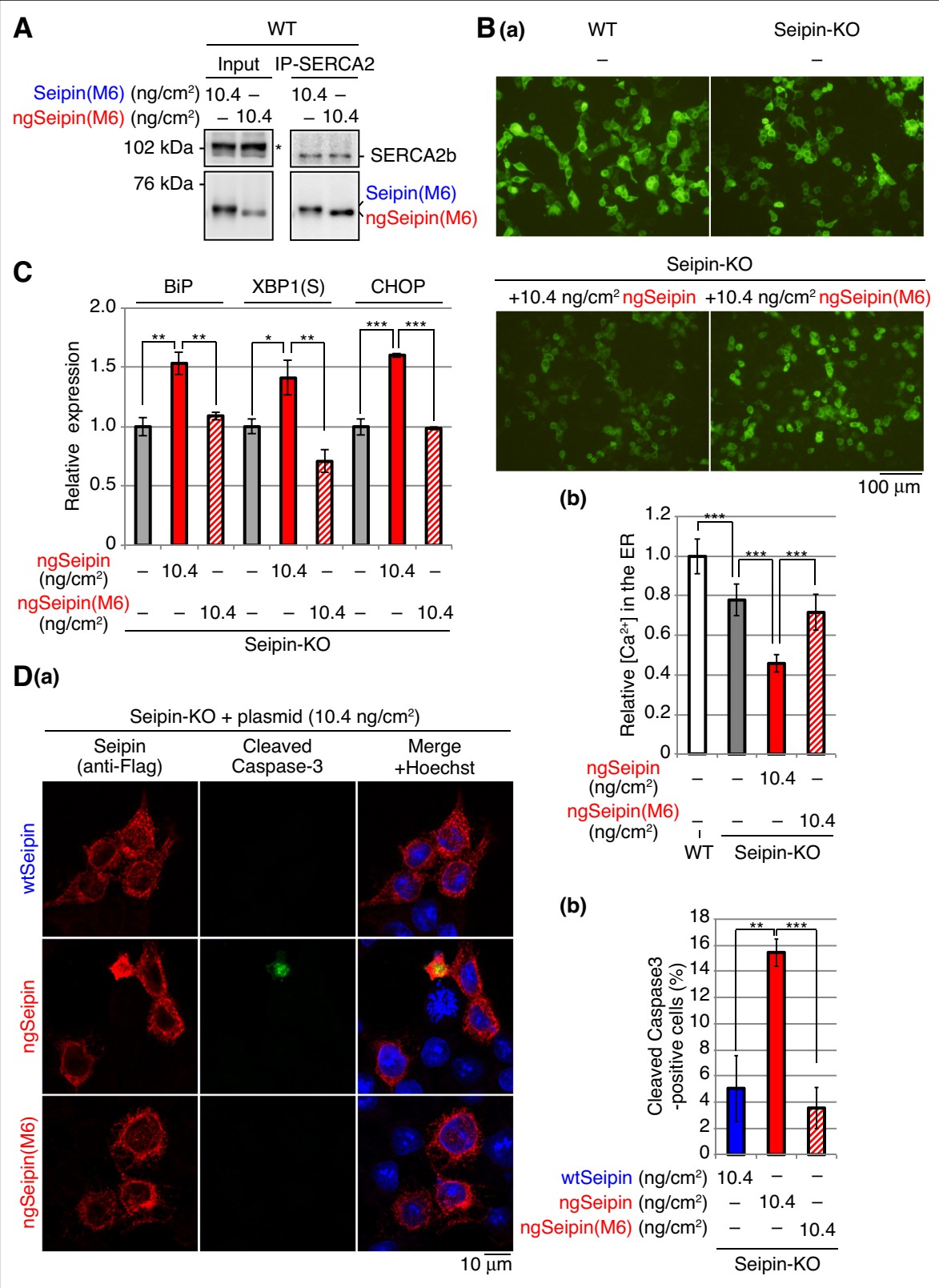

**Figure 5.** Effect of non-glycosylated Seipin (ngSeipin) oligomerization on calcium concentration in the endoplasmic reticulum (ER), ER stress, and apoptosis in Seipin-KO HCT116 cells. (**A**) Cell lysates were prepared from HCT116 WT cells transfected with plasmid (10.4 ng/cm²) to express Myc-tagged Seipin(M6) or ngSeipin(M6), subjected to immunoprecipitation using anti-SERCA2 antibody, and analyzed as in *Figure 1C*. The asterisk denotes a nonspecific band. (**B**) HCT116 WT cells were transfected with plasmid (104 ng/cm²) to express G-CEPIA1er. Seipin-KO cells were transfected

*Figure 5 continued on next page*

*Figure 5 continued*

with plasmid (104 ng/cm²) to express G-CEPIA1er together with or without plasmid (10.4 ng/cm²) to express Myc-tagged ngSeipin or ngSeipin(M6). (**a**) Fluorescent microscopic analysis was conducted. Scale bar: 100 μm. (**b**) Fluorescence intensities were quantified and are expressed as in *Figure 1E(b)* (n = 3). (**C**) Quantitative RT-PCR was conducted in Seipin-KO cells transfected with or without plasmid (10.4 ng/cm²) to express Myc-tagged ngSeipin or ngSeipin(M6) (n = 3), as in *Figure 3B*. (**D**) (**a, b**) Seipin-KO cells transfected with plasmid (10.4 ng/cm²) to express Flag-tagged wild-type Seipin (wtSeipin), ngSeipin, or ngSeipin(M6) were fixed 28 hr later, subjected to immunofluorescence, and analyzed as in *Figure 3C* using 108–118 cells (n = 3). Scale bar: 10 μm. See also *Figure 5—source data 1*.

The online version of this article includes the following source data for figure 5:

**Source data 1.** Raw data related to *Figure 5A*.

double-negative regions increased in a dose (transfection level of ngSeipin$^L$)-dependent manner (*Figure 8B(c)*). It should be noted that strong signals by PLA were observed around double-negative regions when Myc-tagged ngSeipin$^L$ and Flag-tagged ngSeipin$^L$ were co-expressed by transfection (*Figure 4E(a)*, panel 8-II, double-negative regions are surrounded by yellow broken lines).

Importantly, such double-negative regions were hardly observed in Seipin-KO cells expressing Seipin$^L$(M6) or ngSeipin$^L$(M6) by transfection at 10.4 ng/cm² (*Figure 8A(a, c)*, compare bar 8 with bars 9 and 10), but were observed in Seipin-KO cells expressing ngSeipin$^L$(ΔC) to the same extent as in Seipin-KO cells expressing ngSeipin$^L$ (*Figure 8—figure supplement 1A(a, b)*, compare bar 3 with bar 2). Thus, induction of double-negative regions correlates well with aggregation propensity of ngSeipin$^L$ determined by PLA (*Figure 7D*). Of note, double-negative regions were not stained with anti-GAPDH antibody (*Figure 8B*).

We found that this distorted morphology of the ER explained the unexpected decrease in [Ca$^{2+}$] in the ER in Seipin-KO cells expressing Seipin$^L$(ΔN), Seipin$^L$(TM1), and Seipin$^L$(TM2) by transfection (*Figure 6D*, bars 5, 7, and 10). Such cells contained double-negative regions similarly to Seipin-KO cells expressing ngSeipin$^L$ (*Figure 8—figure supplement 1A(a, b)*, compare bar 2 with bars 4, 6, and 8), and the percentage of cells with double-negative regions increased in Seipin-KO cells expressing their respective non-glycosylated version (*Figure 8—figure supplement 1A(a, b)*, bars 5, 7, and 9). It is likely that truncation of the N-terminal region or swapping of the transmembrane domain per se adversely affected the structural maintenance of Seipin, making the Seipin mutants aggregation-prone.

Double-negative regions were also observed in Seipin-KO cells simultaneously expressing mCherry-SERCA2b and mEGFP-ngSeipin$^L$ by transfection in a dose-dependent manner (*Figure 8—figure supplement 2A(a, d)*, bars 2, 4, and 6) but not in Seipin-KO cells simultaneously expressing mCherry-SERCA2b and mEGFP-wtSeipin$^L$, mEGFP-Seipin$^L$(M6), or mEGFP-ngSeipin$^L$(M6) (*Figure 8—figure supplement 2A(a, c, d)*, bars 5, 7, and 8). Importantly, such double-negative regions contained ER-tagBFP co-transfected (*Figure 8—figure supplement 2A(b)*) but not LAMP1-mCherry (*Figure 8—figure supplement 2B*), and may be identical to organized smooth ER whorl structures observed in cells overexpressing oligomeric fluorescent proteins (*Cranfill et al., 2016*). These results indicate that oligomerization-dependent aggregates of ngSeipin$^L$ distorted the shape of the ER, incorporated SERCA2b, and produced double-negative regions in the ER.

## Reversal of the effect of ngSeipin by increase in the level of SERCA2b

We examined whether the increase in the level of SERCA2b compensates for the decrease in [Ca$^{2+}$] in the ER caused by expression of ngSeipin$^L$. Overexpression of SERCA2b (WT) but not inactive mutant SERCA2b (Q108H) (*Miyauchi et al., 2006*) by transfection at 20.8 ng/cm² significantly increased [Ca$^{2+}$] in the ER of WT cells expressing ngSeipin$^L$ by transfection at 10.4 ng/cm² (*Figure 9—figure supplement 1A(a, b)*, compare bar 2 with bars 3 and 4). Lowered [Ca$^{2+}$] in the ER of Seipin-KO cells (*Figure 9—figure supplement 1B*, gray bars) as well as of Seipin-KO cells expressing wtSeipin$^L$ by transfection at 10.4 ng/cm² (*Figure 9—figure supplement 1B*, blue bars), compared with WT cells (*Figure 9—figure supplement 1B*, white bar), was significantly increased by the introduction of SERCA2b in a dose-dependent manner, whereas further lowering of [Ca$^{2+}$] in the ER of Seipin-KO cells expressing ngSeipin$^L$ by transfection at 10.4 ng/cm² (*Figure 9—figure supplement 1B*, red bars) was significantly and slightly increased only by transfection at 20.8 ng/cm² of SERCA2b, indicating the severe inactivation of SERCA2b by ngSeipin$^L$.

Introduction of SERCA2b (transfection at 20.8 ng/cm²) had no effect on aggregation propensity between Myc-tagged ngSeipin$^L$ and Flag-tagged ngSeipin$^L$ determined by PLA (*Figure 9A*) or

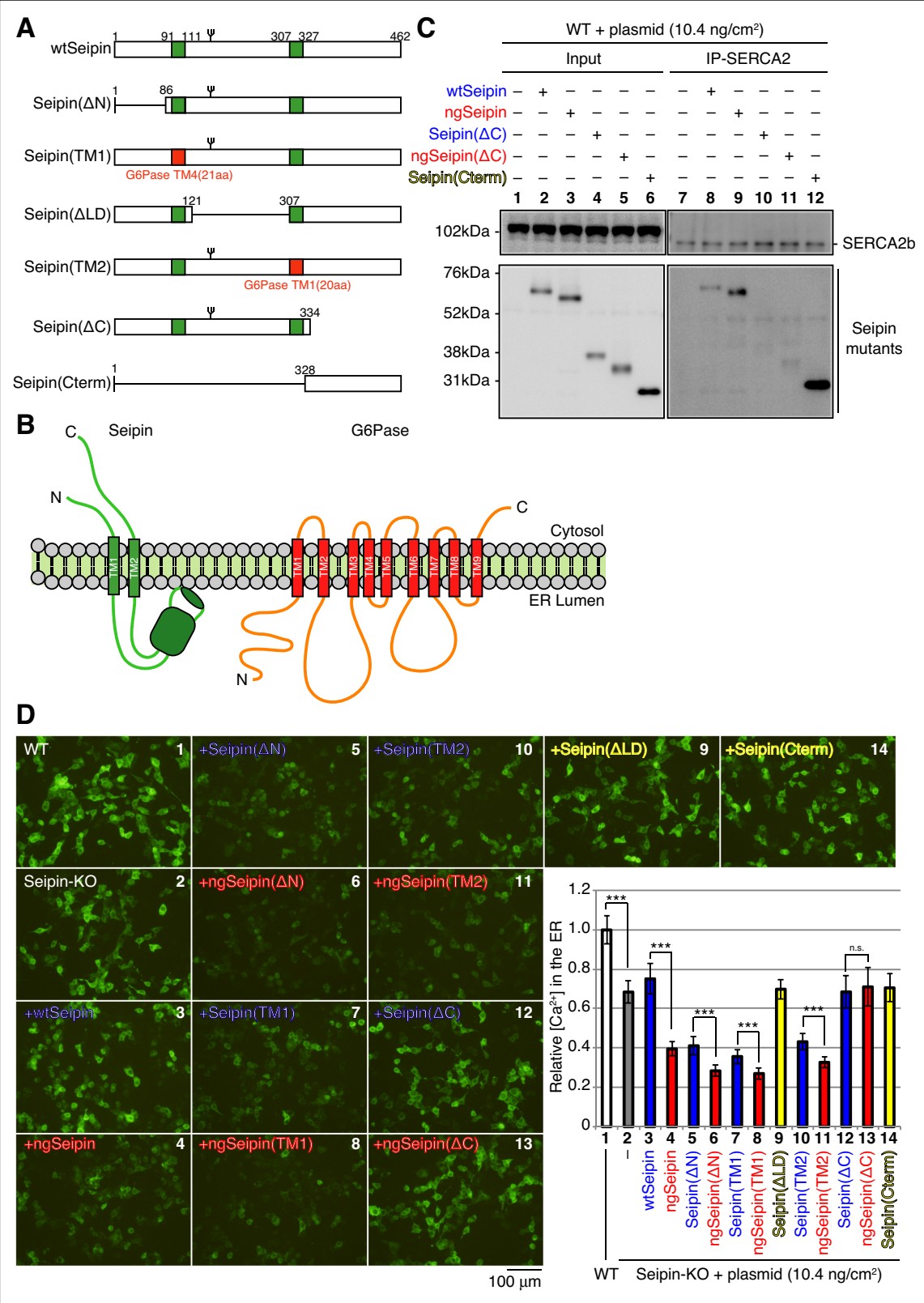

**Figure 6.** Effect of various deletions or replacements of non-glycosylated Seipin (ngSeipin) on interaction with SERCA2b and calcium concentration in the endoplasmic reticulum (ER) of Seipin-KO HCT116 cells. (**A**) Structures of wild-type Seipin (wtSeipin) and various mutant Seipin are schematically shown. The red boxes denote the transmembrane (TM1 and TM4) domains of glucose 6-phosphatase (G6Pase) to swap the TMs of Seipin. (**B**) Structures of Seipin and G6Pase are schematically shown. (**C**) Cell lysates were prepared from HCT116 WT cells transfected with plasmid (10.4 ng/cm²) to

*Figure 6 continued on next page*

*Figure 6 continued*

express Myc-tagged wtSeipin or various mutant Seipin, subjected to immunoprecipitation using anti-SERCA2 antibody, and analyzed as in *Figure 1C*. (**D**) HCT116 WT cells were transfected with plasmid (104 ng/cm$^2$) to express G-CEPIA1er. Seipin-KO cells were transfected with plasmid (104 ng/cm$^2$) to express G-CEPIA1er together with or without plasmid (10.4 ng/cm$^2$) to express Myc-tagged wtSeipin or various mutant Seipin. (a) Fluorescent microscopic analysis of WT or Seipin-KO cells transfected as indicated was conducted. Scale bar: 100 µm. (b) Fluorescence intensities were quantified and are expressed as in *Figure 1E(b)* (n = 3). See also *Figure 6—source data 1*.

The online version of this article includes the following source data and figure supplement(s) for figure 6:

**Source data 1.** Raw data related to *Figure 6C*.

**Figure supplement 1.** Effect of various deletions or replacements of non-glycosylated Seipin (ngSeipin) on interaction with SERCA2b.

**Figure supplement 1—source data 1.** Raw data related to *Figure 6—figure supplement 1*.

ngSeipin$^L$-mediated distortion of ER morphology determined by immunofluorescence (*Figure 9B*). Critically, however, introduction of SERCA2b (transfection at 20.8 ng/cm$^2$) significantly mitigated the ER stress induced in Seipin-KO cells expressing ngSeipin$^L$ by transfection at 10.4 ng/cm$^2$ (*Figure 9C*). Accordingly, the growth rate of WT cells slowed by expression of ngSeipin$^L$ by transfection at 10.4 ng/cm$^2$ was rescued by the introduction of SERCA2b (transfection at 20.8 ng/cm$^2$) (*Figure 9D*). The percentage of apoptotic cells markedly increased by expression of ngSeipin by transfection at 10.4 ng/cm$^2$ was greatly reduced by introduction of SERCA2b by transfection at 20.8 ng/cm$^2$ (*Figure 9E*). We concluded that ngSeipin induces ER stress and apoptosis through the inactivation of SERCA2b.

## Effect of ngSeipin expression in SH-SY5Y cells

We intended to knock-in N152S or S154L mutation in SH-SY5Y cells but failed to do so, possibly due to its slow growth rate with doubling time of ~67 hr (*Feles et al., 2022*) and low transfection efficiency (~15%). Instead, we obtained Seipin-KO cells by CRISPR/Cas9-mediated cleavage at exon 3, in which one nucleotide was deleted from exon 3, causing a frame shift at aa156 (*Figure 10—figure supplement 1A–G*). The absence of Seipin in SH-SY5Y KO-cells was confirmed by immunoblotting (*Figure 10—figure supplement 1H*).

The ratio of expression level of Seipin$^L$ to Seipin$^S$ in SH-SY5Y cells was estimated to be 4 : 9 by quantification of the results of immunoblotting (*Figure 10—figure supplement 1H*). Accordingly, the ratio of amounts of plasmids to express wtSeipin$^L$/ngSeipin$^L$ and wtSeipin$^S$/ngSeipin$^S$ by transfection was set to be 4 : 9. Immunoblotting of cell lysates prepared from SH-SY5Y cells that had been transfected with various amounts of plasmid revealed that 0.62 ng/cm$^2$ and 1.40 ng/cm$^2$ of plasmid to express wtSeipin$^L$ and wtSeipin$^S$, respectively, was the minimum required to clearly detect them. These expression levels were estimated to be ~16% of those of endogenous Seipin$^L$ and Seipin$^S$ (*Figure 10A(c)*). Given a transfection efficiency of ~15%, as estimated by transfecting SH-SY5Y cells with plasmid (104 ng/cm$^2$) to express G-CEPIA1er (*Figure 10—source data 1*), the levels of ngSeipin$^L$ and ngSeipin$^S$ expressed by transfection in Seipin-KO SH-SY5Y cells would be considered comparable to those of endogenous Seipin$^L$ and Seipin$^S$ in SH-SY5Y WT cells.

[Ca$^{2+}$] in the ER in Seipin-KO cells was significantly decreased compared with WT cells (*Figure 10A(a, b)*, compare bar 2 with bar 1). This decrease was rescued by expression of wtSeipin$^L$ and wtSeipin$^S$ by transfection (0.62 ng/cm$^2$ and 1.40 ng/cm$^2$, respectively) but further decreased by expression of ngSeipin$^L$ and ngSeipin$^S$ by transfection (0.62 ng/cm$^2$ and 1.40 ng/cm$^2$, respectively) or by simultaneous expression of wtSeipin$^L$ and ngSeipin$^L$ (0.31 ng/cm$^2$ each) as well as wtSeipin$^S$ and ngSeipin$^S$ (0.70 ng/cm$^2$ each) by transfection (*Figure 10A(a, b)*, compare bars 3, 4, and 5 with bars 1 and 2).

Accordingly, distorted morphology of the ER (occurrence of strongly double-positive and double-weak regions) (*Figure 10B*); induction of ER stress as evidenced by activation of the ERSE (mainly regulated by the ATF6 pathway), UPRE (regulated by the IRE1 and ATF6 pathways), and ATF4 (regulated by the PERK pathway) reporters (*Figure 10C*); and induction of apoptosis (*Figure 10D*) were observed in Seipin-KO SH-SY5Y cells expressing ngSeipin$^L$ and ngSeipin$^S$ (0.62 ng/cm$^2$ and 1.40 ng/cm$^2$, respectively) by transfection (*Figure 10B–D*, bar 5) and Seipin-KO cells simultaneously expressing wtSeipin$^L$ and ngSeipin$^L$ (0.31 ng/cm$^2$ each), as well as wtSeipin$^S$ and ngSeipin$^S$ (0.70 ng/cm$^2$ each) by transfection (*Figure 10B–D*, bar 4), but not Seipin-KO cells expressing wtSeipin$^L$ and wtSeipin$^S$ (0.62 ng/cm$^2$ and 1.40 ng/cm$^2$, respectively) by transfection (*Figure 10B–D*, bar 3). Note that much a higher frequency in the occurrence of apoptotic cells was observed in SH-SY5Y cells than in HCT116 cells

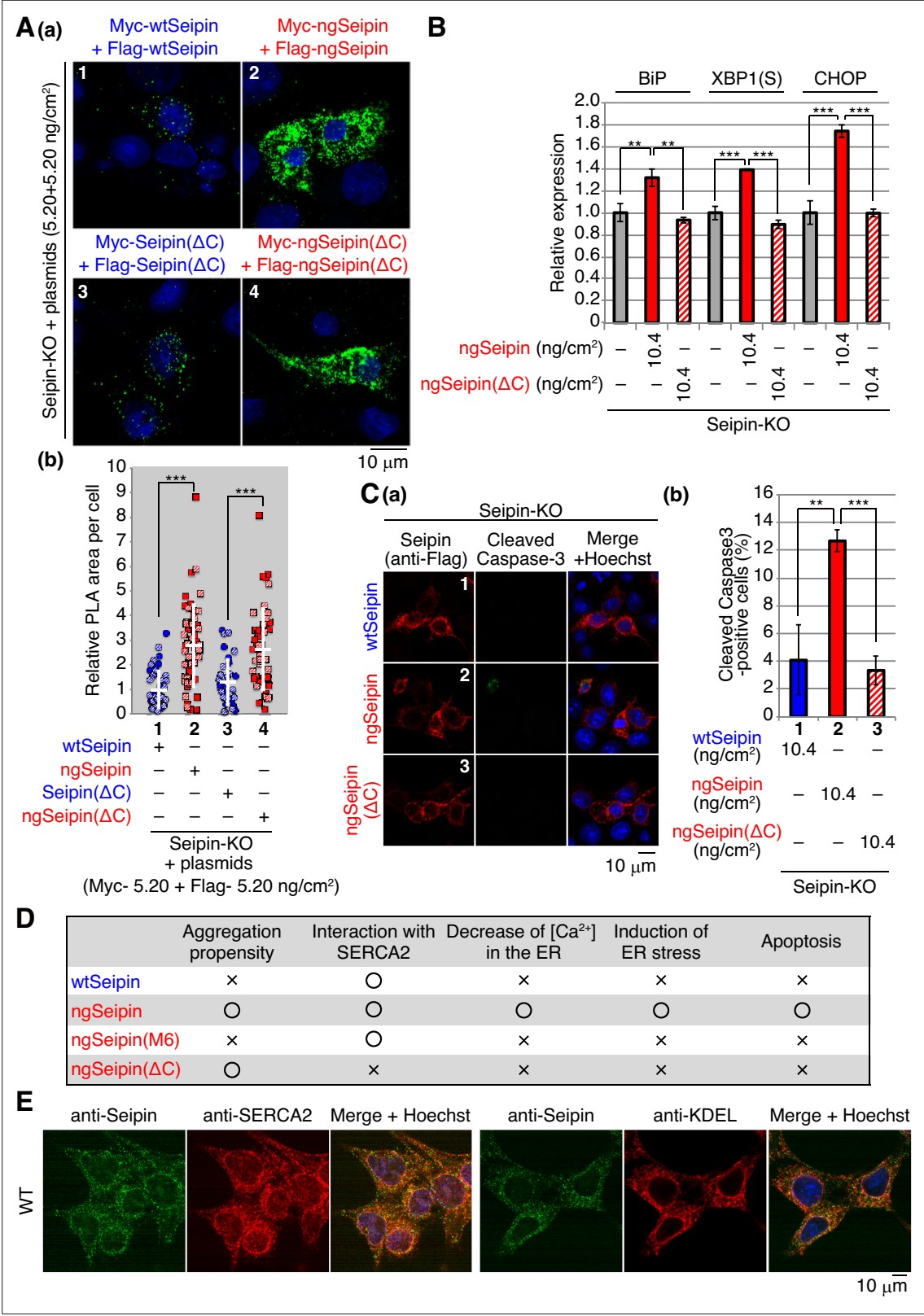

**Figure 7.** Effect of C-terminal deletion of non-glycosylated Seipin (ngSeipin) on aggregation of Seipin, endoplasmic reticulum (ER) stress, and apoptosis in Seipin-KO HCT116 cells. (**A**) (**a, b**) Seipin-KO cells transfected with plasmids (5.20 ng/cm² each) to simultaneously express Myc-tagged and Flag-tagged wild-type Seipin (wtSeipin) or various mutant Seipin as indicated were subjected to proximity ligation assay (PLA) and analyzed as in *Figure 4E* using 40–48 cells (n = 2). Scale bar: 10 µm. (**B**) Quantitative RT-PCR was conducted in Seipin-KO cells transfected with or without plasmid (10.4 ng/cm²) to

*Figure 7 continued on next page*

Figure 7 continued

express Myc-tagged ngSeipin or ngSeipin(ΔC) (n = 3), as in *Figure 3B*. (**C**) (**a, b**) Seipin-KO cells transfected with plasmid (10.4 ng/cm²) to express Flag-tagged wtSeipin, ngSeipin, or ngSeipin(ΔC) were fixed 28 hr later, subjected to immunofluorescence, and analyzed as in *Figure 3C* using117-127 cells (n = 3). Scale bar: 10 μm. (**D**) Phenotypes of wtSeipin, ngSeipin, ngSeipin(M6), and ngSeipin(ΔC) are summarized. (**E**) HCT116 WT cells were analyzed by immunofluorescence using anti-Seipin, anti-SERCA2, and anti-KDEL antibodies with fluorescence microscopy (AiryScan). Scale bar: 10 μm.

(compare *Figure 10D* with *Figure 5D*), suggesting that SH-SY5Y cells are more vulnerable to ER stress than HCT116 cells. We concluded that the expression of ngSeipin at an endogenous protein level induces ER stress and subsequent apoptosis by decreasing [Ca²⁺] in the ER in a dominant manner in SH-SY5Y cells.

## Discussion

Seipin is conserved from yeast to humans, and yeast and fly orthologs are termed Sei1 (Fld1) and dSeipin, respectively. Interestingly, Sei1 and dSeipin do not have potential *N*-glycosylation sites, and dSeipin was shown to function as a dodecamer of non-glycosylated monomer (*Sui et al., 2018*). In contrast, Seipin orthologs in vertebrates have gained one *N*-glycosylation site (Asn¹⁵²Val¹⁵³Ser¹⁵⁴) in the luminal region, and one of the two *N*-acetylglucosamines proximal to Asn¹⁵² (Glc₃Man₉Glc-NAc₂-Asn) was shown to interact with Arg¹⁹⁹ and Gly²⁰⁰ in the same molecule (*Yan et al., 2018*) with reference to Protein Data Bank: 6DS5. This presence of *N*-glycan at the interface of each vertebrate Seipin protomer to the next vertebrate Seipin protomer in the same direction is likely to prevent dodecamer formation and to instead induce undecamer formation (*Figure 4A*). In this sense, non-glycosylated vertebrate Seipin is speculated to function as a dodecamer. Indeed, ngSeipin expressed at an endogenous protein level in Seipin-KO cells (transfection at 0.52 ng/cm², *Figure 3A(c)*; compare lane 4 with lane 1) rescued [Ca²⁺] in the ER, which was mitigated in Seipin-KO cells compared with WT cells (*Figure 3A(a, b)*; compare bar 4 with bars 1 and 2), suggesting that the putative dodecamer of non-glycosylated human Seipin is functional. These results suggest that non-glycosylation of human Seipin itself is not detrimental to the cell.

However, the presence of ngSeipin in Seipin-KO cells at a higher level (transfection at ≥2.60 ng/cm²) causes serious problems, namely, a more profound decrease in [Ca²⁺] in the ER than the presence of the equivalent amount of wtSeipin (*Figure 3A(a, b)*; compare bars 6, 8, and 10 with bars 5, 7, and 9). ngSeipin decreased [Ca²⁺] in the ER in a dominant manner over wtSeipin (*Figure 2B*), consistent with its disease-causing phenotype. The decrease in [Ca²⁺] in the ER in turn induced ER stress and activation of the UPR, leading to apoptosis (*Figure 3B and C*; compare bars 6, 8, and 10 with bars 5, 7, and 9), as ER-localized molecular chaperones require Ca²⁺ for their function; it was indeed shown that Ca²⁺ depletion destabilizes BiP-substrate complexes (*Preissler et al., 2020*). The decrease in [Ca²⁺] in the ER also likely detrimentally affects intracellular signaling and synaptic transmission in the nervous system.

Furthermore, we unraveled the underlying molecular mechanism: ngSeipin expressed at a higher level decreases [Ca²⁺] in the ER by inactivating SERCA2b, for which two independent phenomena are prerequisite, namely, oligomerization-dependent aggregation and C-terminal region-dependent direct association with SERCA2b (*Figure 7D*). Thus, the prevention of oligomerization by mutating the six amino acids in the luminal region required for oligomerization (ngSeipinᴸ(M6), *Figure 4B*) abolished aggregation of ngSeipin (*Figure 4E(a, b)*; compare dot blot 10 with dot blot 8, *Figure 8A(a, c)*; compare bar 10 with bar 8), the ngSeipin-mediated decrease in [Ca²⁺] in the ER (*Figure 5B*), induction of ER stress (*Figure 5C*), and induction of apoptosis (*Figure 5D*).

On the other hand, deletion of the C-terminal region of Seipin (ngSeipinᴸ(ΔC), *Figure 6A*) abolished the ngSeipin-mediated decrease in [Ca²⁺] in the ER (*Figure 6D*, compare bar 13 with bar 4), induction of ER stress (*Figure 7B*), and induction of apoptosis (*Figure 7C*), even though this ngSeipinᴸ(ΔC) exhibited a propensity for aggregation (*Figure 7A(a, b)*; compare dot blot 4 with dot blot 2, *Figure 8—figure supplement 1A(a, b)*; compare bar 3 with bar 2), indicating that the aggregation of ngSeipin is not sufficient to induce ER stress. These results in turn mean that ngSeipin expressed at a higher level cannot maintain the putative dodecamer structure, aggregates and distorts the ER morphology, and then incorporates SERCA2b into aggregates via direct interaction of its C-terminus, resulting in inactivation of SERCA2b.

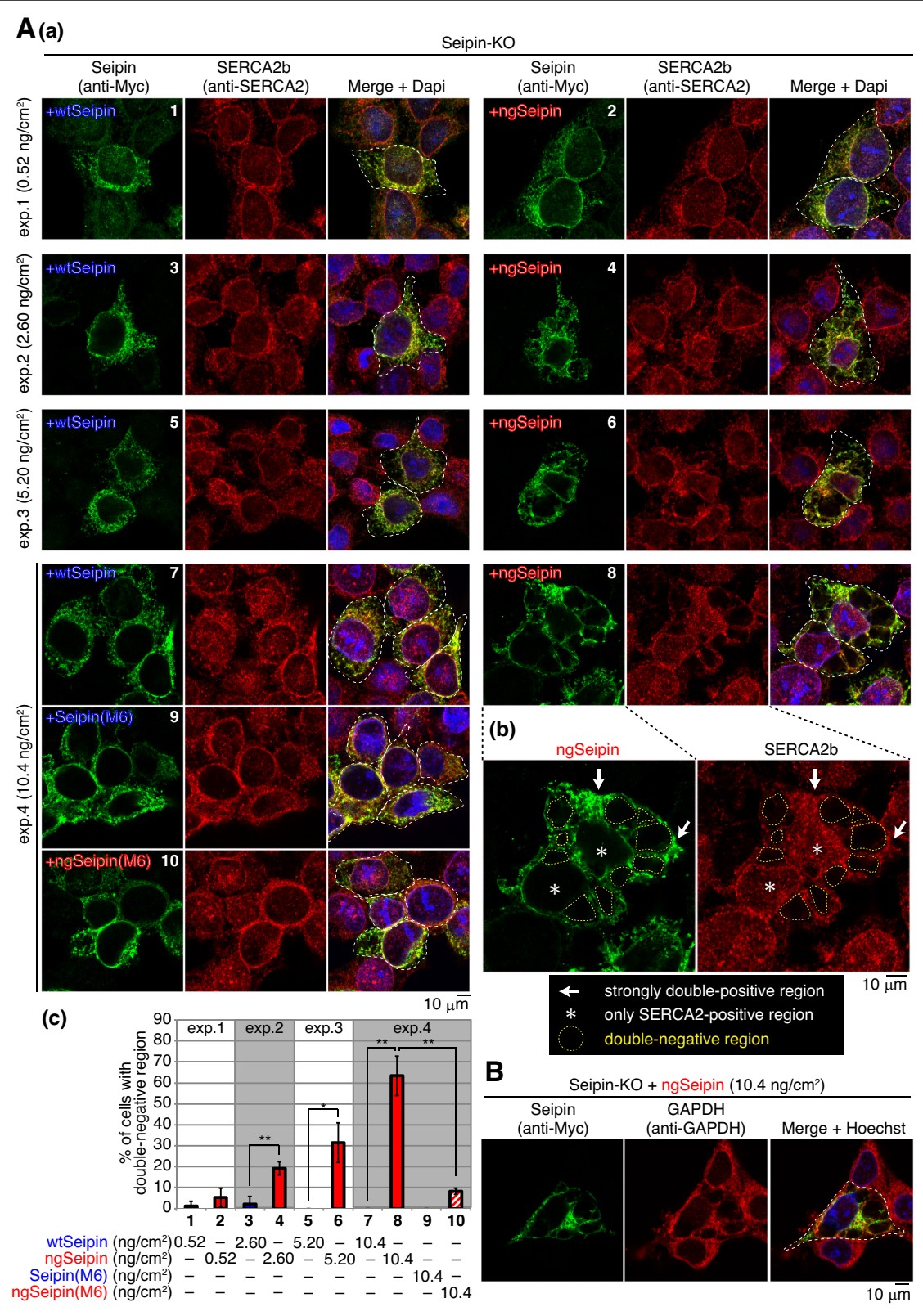

**A (a)**

**Figure 8.** Effect of non-glycosylated Seipin (ngSeipin) expression on morphology of the endoplasmic reticulum (ER) and localization of SERCA2b. (**A**) (**a**) Seipin-KO cells transfected with the indicated amounts of plasmid to express Myc-tagged wild-type Seipin (wtSeipin), ngSeipin, Seipin(M6), or ngSeipin(M6) were analyzed by immunofluorescence using anti-Myc and anti-SERCA2 antibodies with fluorescence microscopy (AiryScan). Transfected cells are surrounded by white broken lines. Scale bar: 10 μm. (**b**) Immunofluorescence images of Seipin-KO cells expressing ngSeipin by transfection

*Figure 8 continued on next page*

*Figure 8 continued*

(10.4 ng/cm²) are enlarged. Strongly double-positive regions and SERCA2-positive-only regions are indicated by white arrows and asterisks, respectively. Double-negative regions are surrounded by yellow broken lines. (**c**) Percentages of cells containing double-negative regions were quantified and are shown (n = 3, total 70–132 cells analyzed). (**B**) Seipin-KO cells transfected with plasmid (10.4 ng/cm²) to express Myc-tagged ngSeipin were analyzed by immunofluorescence using anti-Myc and anti-GAPDH antibodies with fluorescence microscopy (AiryScan). Scale bar: 10 μm.

The online version of this article includes the following figure supplement(s) for figure 8:

**Figure supplement 1.** Effect of various deletions or replacements of Seipin on morphology of the endoplasmic reticulum (ER) and localization of SERCA2b.

**Figure supplement 2.** Effect of mEGFP-ngSeipin expression on morphology of the endoplasmic reticulum (ER) and localization of SERCA2b.

Of note, SH-SY5Y cells appears to be more vulnerable to the effect of ngSeipin expression than HCT116 cells; expression of ngSeipin by transfection at an endogenous protein level was sufficient to cause a decrease in [Ca²⁺] in the ER (*Figure 10A*), distortion of the ER morphology (*Figure 10B*), induction of ER stress (*Figure 10C*), and induction of apoptosis (*Figure 10D*), in a dominant manner.

Importantly, the increase in the level of SERCA2b mitigated ngSeipin-mediated induction of ER stress (*Figure 9C*) and subsequent apoptosis (*Figure 9E*) even though ngSeipin was still severely aggregated (*Figure 9A and B*). This raises the intriguing possibility that a potent SERCA2b activator could be used as a therapeutic for Seipinopathy and other ER stress-associated neurodegenerative diseases. In this connection, it was reported that high levels of ER stress markers were observed in motor neurons derived from patients with amyotrophic lateral sclerosis, a severe motor neuron disease, carrying the A4V mutation in superoxide dismutase 1 (*Kiskinis et al., 2014*; *Wainger et al., 2014*); and that oral administration of sodium phenylbutyrate and taurursodiol, chemical chaperones that mitigate ER stress, significantly slowed functional decline in patients with amyotrophic lateral sclerosis (*Paganoni et al., 2021*; *Paganoni et al., 2020*). These findings highlight the increasing importance of ER stress in understanding the development of certain neurodegenerative diseases.

# Materials and methods

**Key resources table**

| Reagent type (species) or resource | Designation | Source or reference | Identifiers | Additional information |
|---|---|---|---|---|
| | | | | Parental cell line has been authenticated using STR profiling |
| Cell line (*Homo sapiens*) | Colorectal carcinoma | ATCC | HCT116 | All cell lines have been tested negative for mycoplasma |
| | | | | Parental cell line has been authenticated using STR profiling |
| Cell line (*H. sapiens*) | Neuroblastoma | ATCC | SH-SY5Y | All cell lines have been tested negative for mycoplasma |
| Recombinant DNA reagent | pCMV-Myc | Clontech | | |
| Recombinant DNA reagent | pCMV G-CEPIA1er | Addgene | | |
| Recombinant DNA reagent | pGP-CMV-GCaMP6f | Addgene | | |
| Recombinant DNA reagent | px330-U6-Chimeric_BB-CBh-hSpCas9 | Addgene | | |
| Antibody | Anti-BSCL2(Seipin) (rabbit monoclonal) | Cell Signaling | Cat# 23846 | WB (1:1000) Immunostaining (1:1000) |
| Antibody | Anti-SERCA2 (mouse monoclonal) | Santa Cruz | Cat# sc-376235 | WB (1:500) IP (5 μl) Immunostaining (1:250) |

*Continued on next page*

*Continued*

| Reagent type (species) or resource | Designation | Source or reference | Identifiers | Additional information |
|---|---|---|---|---|
| Antibody | Anti-Myc-tag mAb-HRP-DirecT (mouse monoclonal) | MBL | Code M047-7 | WB (1:1000) |
| Antibody | Anti-GFP (rabbit polyclonal) | MBL | Code 598 | WB (1:1000) |
| Antibody | Anti-β-actin (mouse monoclonal) | Wako | Cat# 017-24573 | WB (1:2000) |
| Antibody | Anti-GAPDH peroxidase conjugated (mouse monoclonal) | Wako | Cat# 015-25473 | WB (1:5000) |
| Antibody | Anti-GAPDH (mouse monoclonal) | Wako | Cat# 014-25524 | Immunostaining (1:1000) |
| Antibody | Anti-KDEL (mouse monoclonal) | MBL | Code M181-3 | Immunostaining (1:2000) |
| Antibody | Anti-Myc-tag (rabbit polyclonal) | MBL | Code 562 | PLA (1:250) Immunostaining (1:100) |
| Antibody | Anti-Flag M2 (mouse monoclonal) | Sigma | Cat# F3165 | PLA (1:250) Immunostaining (1:500) |
| Antibody | Anti-Myc tag (mouse monoclonal) | Wako | Cat# 017-21871 | Immunostaining (1:100) |
| Antibody | Anti-cleaved Caspase-3 (rabbit polyclonal) | Cell Signaling | Cat# 9661 | Immunostaining (1:800) |
| Antibody | Alexa488-conjugated anti-rabbit secondary antibody (goat polyclonal) | Invitrogen | Cat# A-11008 | Immunostaining (1:1000) |
| Antibody | Alexa568-conjugated anti-mouse secondary antibody (goat polyclonal) | Invitrogen | Cat# A-11004 | Immunostaining (1:1000) |

## Statistics

Statistical analysis was conducted using Student's *t*-test, with probability expressed as $*p<0.05$, $**p<0.01$, and $***p<0.001$ for all figures. n.s. denotes not significant.

## Construction of plasmids

Recombinant DNA techniques were performed according to standard procedures (*Sambrook et al., 1989*) and the integrity of all constructed plasmids was confirmed by extensive sequencing analyses. All new constructs are available upon request.

pCMV-Myc (Clontech) was used to express Seipin and SERCA2b tagged with Myc at the N-terminus. Site-directed mutagenesis was carried out using DpnI. Plasmids to express Flag-tagged and HA-tagged proteins were created by changing the Myc-coding sequence in pCMV-Myc expression plasmids to the intended tag-coding sequence using inverse PCR and NE Builder HiFi Assembly (New England Biolabs), respectively.

To create constructs to express various forms of Seipin fused with mEGFP at the N-terminus, mEGFP fragment amplified from an mEGFP-KDEL expression vector (*George et al., 2021*) was inserted between Myc and the Seipin-coding sequence. To create a construct to express SERCA2b fused with mCherry at the N-terminus, mCherry fragment amplified from pmCherry-N1 expression vector (TAKARA) was inserted between Myc and the SERCA2b-coding sequence. The construct to express ER-TagBFP was created by replacing the Myc sequence of pCMV-Myc with the TagBFP sequence, which was fused with the calreticulin signal sequence at the N-terminus and the KDEL (ER retention signal) sequence at the C-terminus. The pLAMP1-mCherry vector was obtained from Amy Palmer (Addgene plasmid #45147).

## Cell culture and transfection

HCT116 cells (ATCC CCL-247) and SH-SY5Y cells (ATCC CRL-2266) were cultured in Dulbecco's modified Eagle's medium (glucose 4.5 g/l) supplemented with 10% fetal bovine serum, 2 mM glutamine,

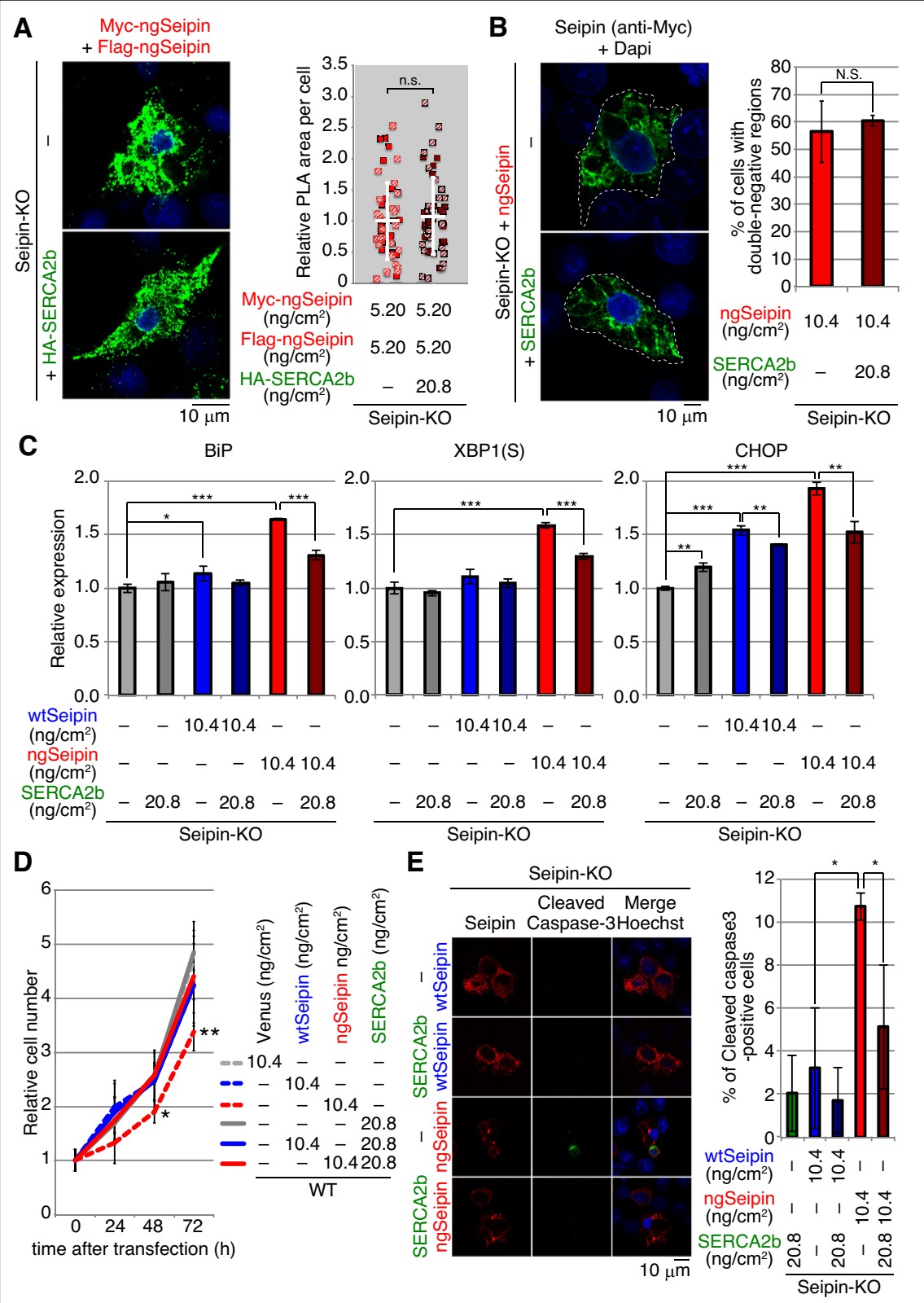

**Figure 9.** Effect of introduction of SERCA2b on aggregation of non-glycosylated Seipin (ngSeipin), endoplasmic reticulum (ER) stress, growth, and apoptosis in ngSeipin-expressing HCT116 cells. (**A**) Seipin-KO cells transfected with plasmids (5.20 ng/cm² each) to simultaneously express Myc-tagged and Flag-tagged ngSeipin together with or without plasmid (20.8 ng/cm²) to express HA-tagged SERCA2b were subjected to proximity ligation assay (PLA) and analyzed as in *Figure 4E* using 49–50 cells (n = 2). Scale bar: 10 μm. (**B**) Seipin-KO cells transfected with plasmid (10.4 ng/cm²) to express

*Figure 9 continued on next page*

*Figure 9 continued*

Myc-tagged ngSeipin together with or without plasmid (20.8 ng/cm$^2$) to express HA-tagged SERCA2b were analyzed by immunofluorescence using anti-Myc antibody with fluorescence microscopy (AiryScan). Percentage of cells containing double-negative regions was determined and shown (n = 3, total 110–112 cells analyzed). (**C**) Quantitative RT-PCR was conducted in Seipin-KO cells transfected with or without plasmid (10.4 ng/cm$^2$) to express Myc-tagged wild-type Seipin (wtSeipin) or ngSeipin together with or without plasmid (20.8 ng/cm$^2$) to express Myc-tagged SERCA2b (n = 3), as in *Figure 3B*. (**D**) Growth rates of HCT116 WT cells transfected with plasmid (10.4 ng/cm$^2$) to express Venus, Myc-tagged wtSeipin or ngSeipin together with or without plasmid (20.8 ng/cm$^2$) to express Myc-tagged SERCA2b were determined by counting cell number every 24 hr (n = 4). Cell number at the time of transfection is set as 1. (**E**) Seipin-KO cells transfected with plasmid (10.4 ng/cm$^2$) to express Flag-tagged wtSeipin or ngSeipin together with or without plasmid (20.8 ng/cm$^2$) to express Myc-tagged SERCA2b were fixed 28 hr later, subjected to immunofluorescence, and analyzed as in *Figure 3C* using 97–125 cells (n = 3). Scale bar: 10 µm.

The online version of this article includes the following figure supplement(s) for figure 9:

**Figure supplement 1.** Effect of introduction of SERCA2b on calcium concentration in the endoplasmic reticulum (ER) of non-glycosylated Seipin (ngSeipin)-expressing HCT116 cells.

and antibiotics (100 U/ml penicillin and 100 µg/ml streptomycin) at 37°C in a humidified 5% CO$_2$/95% air atmosphere. Transfection was performed using polyethylenimine Max (Polyscience) for HCT116 cells and Lipofectamine-LTX (Thermo Fisher Scientific) for SH-SY5Y cells according to the manufacturer's instructions.

## Immunoblotting

Cells cultured in a 6-well plate or 3.5 cm dish were harvested with a rubber policeman and collected by centrifugation at 5000 rpm for 2 min. Cell pellets were lysed in 200 µl of SDS sample buffer (50 mM Tris/HCl, pH 6.8, containing 100 mM dithiothreitol, 2% SDS, and 10% glycerol) containing protease inhibitor cocktail (Nacalai Tesque) and 10 µM MG132. Immunoblotting analysis was carried out according to the standard procedure (*Sambrook et al., 1989*). Chemiluminescence obtained using Western Blotting Luminol Reagent (Santa Cruz Biotechnology) was detected using an LAS-3000mini LuminoImage analyzer (Fuji Film). EndoH was obtained from Calbiochem.

## Immunoprecipitation

HCT116 cells cultured in a 6-well plate were washed with PBS, lysed in 300 µl of high salt buffer (50 mM Tris/Cl, pH 8.0, containing 1% NP-40 and 150 mM NaCl) for 10 min on ice, and clarified by centrifugation at 17,800 × *g* for 10 min at 4°C. Resulting supernatant was subjected to immunoprecipitation using anti-SERCA2 antibody and protein A-coupled Sepharose beads (GE Healthcare). Beads were washed twice with high salt buffer, washed with PBS, and boiled for 5 min in SDS sample buffer.

## Immunofluorescence

Cells grown on coverslips were transiently transfected with plasmid to express Myc- or Flag-tagged protein. After 28 hr cells were fixed, permeabilized with methanol at –30°C for 6.5 min, incubated at 37°C for 2 hr with primary antibody (mouse anti-Flag monoclonal, mouse anti-Myc monoclonal, and rabbit anti-Cleaved Caspase-3 polyclonal antibodies for apoptosis; rabbit anti-Seipin monoclonal, rabbit anti-Myc polyclonal, mouse anti-SERCA2 monoclonal, mouse anti-KDEL monoclonal, and mouse anti-GAPDH monoclonal antibodies for ER morphology), and then with Alexa 488-conjugated anti-rabbit secondary antibody and Alexa 568-conjugated anti-mouse secondary antibody at 37°C for 1 hr. Coverslips were mounted with Prolong Gold or Prolong Grass Antifade Mountants (both from Thermo Fisher Scientific) containing 5 µg/ml Hoechst 33342 or 50 µg/ml Dapi. Images were acquired using an LSM 880 with AiryScan and Zen/Zen2.6 acquisition software (both from Carl Zeiss).

## Live-cell imaging

Cells grown on a glass-bottom dish were transiently transfected with plasmids to express mEGFP-Seipin$^L$ or mEGFP-ngSeipin$^L$, mCherry-SERCA2b or LAMP1-mCherry, and ER-TagBFP. After 28 hr, the medium was changed to fresh DMEM without phenol-red, and live-cell images were acquired using an LSM 880 with AiryScan and Zen/Zen2.6 acquisition software (both from Carl Zeiss).

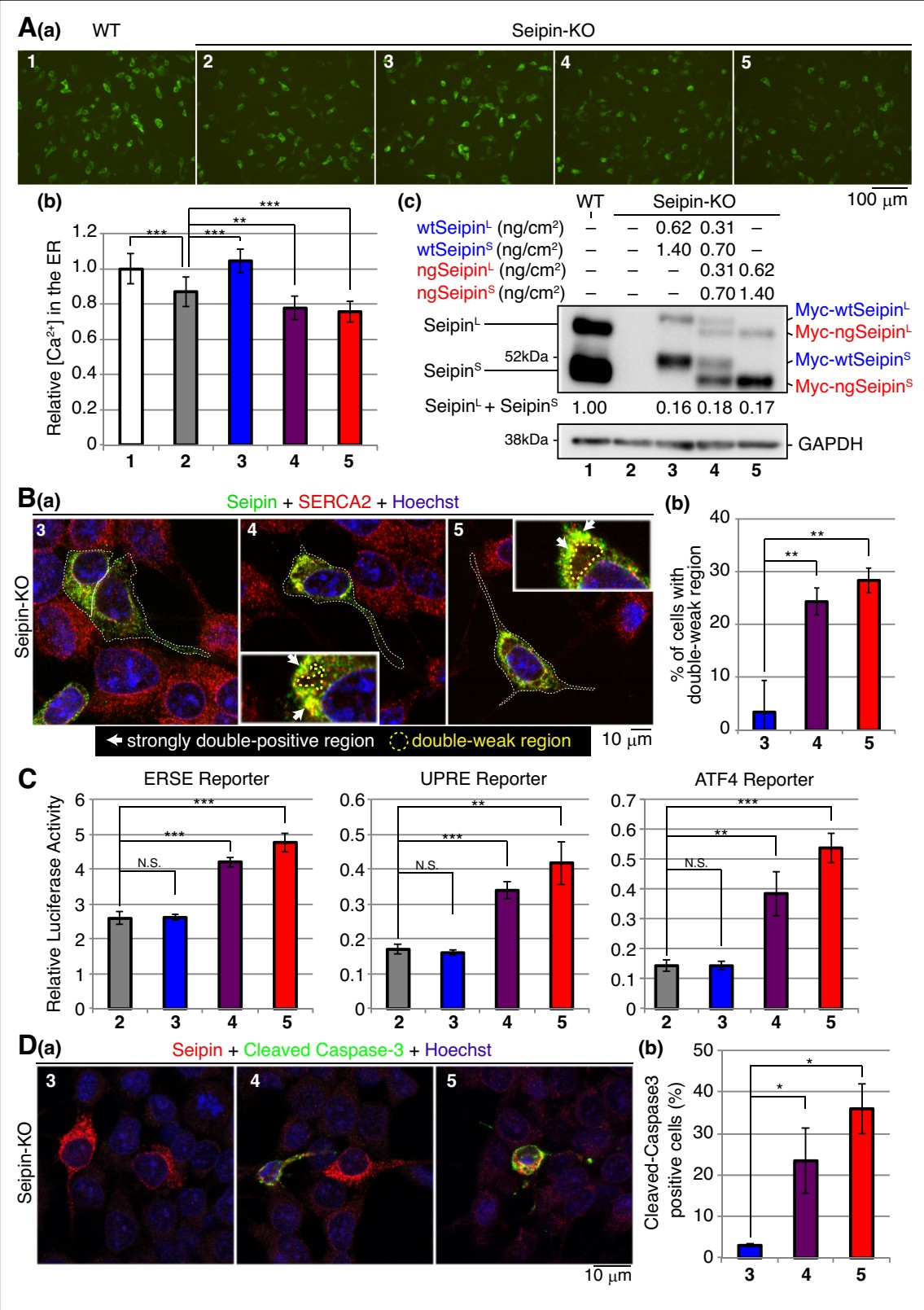

**Figure 10.** Effect of non-glycosylated Seipin (ngSeipin) expression on calcium concentration in the endoplasmic reticulum (ER), morphology of the ER, ER stress, and apoptosis in Seipin-KO SH-SY5Y cells. (**A**) SH-SY5Y WT cells were transfected with plasmid (104 ng/cm²) to express G-CEPIA1er. Seipin-KO cells were transfected with plasmid (104 ng/cm²) to express G-CEPIA1er together with or without the indicated amounts of plasmid to express Myc-tagged wtSeipin$^L$ plus wtSeipin$^S$, ngSeipin$^L$ plus ngSeipin$^S$, or both. (**a**) Fluorescence microscopic analysis of WT cells and Seipin-KO cells

*Figure 10 continued*

transfected as indicated was conducted. Scale bar: 100 µm. (**b**) Fluorescence intensities were quantified and are expressed as in *Figure 1E(b)* (n = 3). (**c**) Cell lysates were prepared from the indicated cells and analyzed by immunoblotting using anti-Seipin and anti-GAPDH antibodies. Quantified data are shown between blots of Seipin and GAPDH. (**B**) (**a**) Seipin-KO cells transfected with the indicated amounts of plasmid to express Myc-tagged wtSeipin$^L$ plus wtSeipin$^S$, ngSeipin$^L$ plus ngSeipin$^S$, or both as in (**A**) were analyzed by immunofluorescence using anti-Seipin and anti-SERCA2 antibodies with fluorescence microscopy (AiryScan). Transfected cells are surrounded by white broken lines. Scale bar: 10 µm. Strongly double-positive regions are indicated by white arrows. Double-weak regions are surrounded by yellow broken lines. (**c**) Percentages of cells containing double-weak regions were quantified and are shown (n = 3, total 90–98 cells analyzed). (**C**) Seipin-KO cells were transfected with the indicated amounts of plasmid to express Myc-tagged wtSeipin$^L$ plus wtSeipin$^S$, ngSeipin$^L$ plus ngSeipin$^S$, or both as in (**A**) together with the ERSE, UPRE, or ATF4 reporter (104 ng/cm$^2$) and the reference plasmid pRL-SV40 (10.4 ng/cm$^2$). Cell lysates were prepared 28 hr later and luciferase activities were determined (n = 3). (**D**) Seipin-KO cells transfected with the indicated amounts of plasmid to express Myc-tagged wtSeipin$^L$ plus wtSeipin$^S$, ngSeipin$^L$ plus ngSeipin$^S$, or both as in (**A**) were fixed 28 hr later, subjected to immunofluorescence, and analyzed as in *Figure 3C*. Scale bar: 10 µm. (**b**) Number of Myc-tagged Seipin (red) and cleaved Caspase-3 (green) double-positive cells was counted in 100–110 cells obtained from three independent experiments and shown as a percentage. See also *Figure 10—source data 1*.

The online version of this article includes the following source data and figure supplement(s) for figure 10:

**Source data 1.** Raw data related to *Figure 10A*.

**Figure supplement 1.** Construction of Seipin-KO SH-SY5Y cells.

**Figure supplement 1—source data 1.** Raw data related to *Figure 10—figure supplement 1B, E, and H*.

## Detection of [Ca$^{2+}$] in the ER or cytosol

[Ca$^{2+}$] in the ER or cytosol was determined using G-CEPIA1er or GCaMP6f. Then 28 hr after transfection, Ca$^{2+}$ imaging was performed with a fluorescence stereomicroscope (Olympus IX-71-22TFL/PH) and acquisition software (DP Controller 1.2.1.108). Fluorescence intensities were measured using ImageJ (https://imagej.nih.gov/ij/). After extracting the green channel, subtracting the background (rolling ball radius: 50.0 pixels), and applying threshold, average gray value of whole cells in each image was determined. Bradykinin, 4CmC, and CDN1163 were obtained from Abcam, Tokyo Chemical Industry, and Sigma-Aldrich, respectively.

## CRISPR/Cas9 method to generate KO cell lines

PuroR fragment amplified by PCR from DT-A-pA-loxP-PGK-Puro-pA-loxP (*Ninagawa et al., 2014*) was inserted into the PciI site of px330-U6-Chimeric_BB-CBh-hSpCas9 (Addgene) to create px330-PuroR.

To construct Seipin-KO HCT116 cells, the DNA oligonucleotides 5′-CACCGCTCTCACTTTCCGCCATTAG-3′ and 5′-AAACCTAATGGCGGAAAGTGAGAGC-3′, and 5′-CACCGGGGAGTGGGAAAGCTTGCTA-3′ and 5′-AAACTAGCAAGCTTTCCCACTCCCC-3′ to express gRNA for cleavage at exon 2 and the 3′UTR-non-coding region, respectively, of the *BSCL2/Seipin* gene were annealed and inserted into the BbsI site of px330-PuroR separately. HCT116 cells were co-transfected with these two plasmids using polyethylenimine Max and screened for puromycin (0.5 µg/ml) resistance.

To construct Seipin-KO SH-SY5Y cells, the DNA oligonucleotides 5′-CACCCAATGTCTCGCTGACTAA-3′ and 5′-AAACTTAGTCAGCGAGACATTG-3′ to express gRNA for cleavage at exon 3 of the *BSCL2/Seipin* gene were annealed and inserted into the BbsI site of px330-PuroR. SH-SY5Y cells were transfected with this plasmid by electroporation using a Microporator (Digital Bio) with three pulses at 1100 V for 20 ms, and screened for puromycin (0.5 µg/ml) resistance.

## Genomic PCR

Non-homologous end joining in HCT116 cells was confirmed by genomic PCR using KOD-FX Neo (TOYOBO) and a pair of primers, all del Fw and all del Rv, as well as inside Fw and inside Rv (*Supplementary file 1*). Non-homologous end joining in SH-SY5Y cells was confirmed by genomic PCR using KOD-FX Neo and a pair of primers, inside Fw and inside Rv.

## RT-PCR

Total RNA prepared from cultured cells (~5 × 10$^6$ cells) by the acid guanidinium/phenol/chloroform method using ISOGEN (Nippon Gene) was converted to cDNA using Moloney murine leukemia virus reverse transcription (Invitrogen) and oligo-dT primers. A part of the cDNA sequence of Seipin and GAPDH was amplified using PrimeSTAR GXL DNA polymerase (Takara Bio) and pairs of primers,

namely, Seipin cDNA Fw and Seipin cDNA Rv, and GAPDH cDNA Fw and GAPDH cDNA Rv, respectively, from HCT116 cells; Seipin cDNA Fw2 and Seipin cDNA Rv2 from SH-SY5Y cells (*Supplementary file 1*).

## Quantitative RT-PCR

Total RNA extracted as above was subjected to quantitative RT-PCR analysis using the SYBR Green method (Applied Biosystems) and a pair of primers, namely, qBiP Fw and qBiP Rv for *BiP* mRNA, qXBP1 Fw and qXBP1 Rv for spliced *XBP1* mRNA, qCHOP Fw and qCHOP Rv for *CHOP* mRNA, qGAPDH Fw and qGAPDH Rv for *GAPDH* mRNA, qSERCA1 Fw and qSERCA1 Rv for *SERCA1* mRNA, qSERCA2 Fw and qSERCA2 Rv for *SERCA2* mRNA, qSERCA3 Fw and qSERCA3 Rv for *SERCA3* mRNA, qSERCA2abc Fw and qSERCA2a Rv for *SERCA2a* mRNA, qSERCA2abc Fw and qSERCA2b Rv for *SERCA2b* mRNA, and qSERCA2abc Fw and qSERCA2c Rv for *SERCA2c* mRNA, qSeipin Fw and qSeipin Rv for Seipin mRNA, qRyR1 Fw and qRyR1 Rv for *RyR1* mRNA, qIP3R1 Fw and qIP3R1 Rv for *IP3R1* mRNA, qIP3R2 Fw and qIP3R2 Rv for *IP3R2* mRNA, and qIP3R3 Fw and qIP3R3 Rv for *IP3R3* mRNA, (*Supplementary file 1*). A total of 200, 2000, and 20,000 molecules of plasmid carrying Seipin, GAPDH, SERCA1, SERCA2, SERCA3, SERCA2a, SERCA2b, SERCA2c, RyR1, IP3R1, IP3R2, or IP3R3 were used as standards.

## Proximity ligation assay

For PLA, cells grown on a glass-bottom dish were transiently transfected with plasmids to express Myc-tagged and Flag-tagged proteins simultaneously. After 24 hr, cells were fixed and permeabilized with methanol at –30°C for 6.5 min. PLA was performed using Duolink in situ Starter Set GREEN (Sigma-Aldrich) according to manufacturer's instructions using rabbit anti-Myc polyclonal and mouse anti-Flag monoclonal antibodies.

## Determination of cell growth rate

Cells transfected with various plasmids were treated with trypsin, and equal amounts of detached cells were plated to four dishes each. Cell numbers were counted 0, 24, 48, and 72 hr later.

## Reporter assay

SH-SY5Y cells cultured in a 24-well plate were washed with PBS and lysed in Luciferase Assay Lysis Buffer (Toyo Bnet). Luciferase activities were determined using PicaGene Dual-luciferase reporter assay reagent (Toyo Bnet). Relative luciferase activity was defined as the ratio of firefly luciferase activity to renilla luciferase activity. pGL3-GRP78(–132)-luc carrying human BiP promoter (*Yoshida et al., 1998*) is called the ERSE reporter, whereas p5xUPRE-GL3 identical to p5xATF6GL3 (*Wang et al., 2000*) is called the UPRE reporter (*Yoshida et al., 2001*). To create pdCMV-murineATF4-luc2, termed the ATF4 reporter, the –261 to +124 region of murine *ATF4* gene (A of the initiation codon in ATF4 ORF set as +1) was amplified by RT-PCR and inserted into the HindIII site of pGL4.23 (Promega), in which the promoter sequence had been replaced with pdCMV(Δ1–388), a truncated pCMV promoter.

## Acknowledgements

We declare no competing financial interests. We thank Ms. Kaoru Miyagawa for her technical and secretarial assistance. This work was financially supported in part by grants from MEXT, Japan (19K06658 to TI, 18K06216 to SN, 17H06419 to KM) and AMED-CREST, Japan (20gm1410005 to KM).

## Additional information

### Funding

| Funder | Grant reference number | Author |
| --- | --- | --- |
| AMED-CREST, Japan | 20gm1410005 | Kazutoshi Mori |

The funders had no role in study design, data collection and interpretation, or the decision to submit the work for publication.

## Author contributions
Shunsuke Saito, Tokiro Ishikawa, Satoshi Ninagawa, Tetsuya Okada, Investigation; Kazutoshi Mori, Conceptualization, Supervision, Funding acquisition, Writing – original draft, Writing – review and editing

## Author ORCIDs
Shunsuke Saito http://orcid.org/0000-0002-1331-811X
Tokiro Ishikawa http://orcid.org/0000-0003-1718-6764
Satoshi Ninagawa http://orcid.org/0000-0002-8005-4716
Tetsuya Okada http://orcid.org/0000-0002-2513-1301
Kazutoshi Mori http://orcid.org/0000-0001-7378-4019

## Decision letter and Author response
Decision letter https://doi.org/10.7554/eLife.74805.sa1
Author response https://doi.org/10.7554/eLife.74805.sa2

---

## Additional files

### Supplementary files
• MDAR checklist

• Supplementary file 1. Names and sequences of various primers used.

### Data availability
All data generated or analysed during this study are included in the manuscript and supporting file.

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
