## [Editor Report]

Seipin is a multifunctional endoplasmic reticulum-localized protein associated with seemingly unrelated human diseases. Here, the authors establish a correlation between the expression of a particular mutant form of Seipin associated in humans with motor neuron disease and altered intracellular calcium dynamics and allied proteotoxic stress. The article is noted for the clues it provides into how these cellular defects arise and for offering a plausible, but yet unproven hypothesis for the cellular pathology that may account for the human disease phenotype.

---

## [Decision Letter]

**Decision letter after peer review:**

Thank you for submitting your article "Non-glycosylated Seipin to Cause a Motor Neuron Disease Induces ER stress and Apoptosis by Inactivating the ER Calcium Pump SERCA2b" for consideration by *eLife*. Your article has been reviewed by 3 peer reviewers, and the evaluation has been overseen by a Reviewing Editor and Suzanne Pfeffer as the Senior Editor. The reviewers have opted to remain anonymous.

Essential revisions:

1) The reviewers were divided in their perception of the significance attributed to the changes in SERCA activity and calcium metabolism observed in cells expressing wild type and ngSeipin. However, in consultation a consensus emerged whereby it was deemed essential that the perturbations arising from ectopic expression of ngSeipin were also examined in a system mimicking the genetics of the diseases namely a mutant endogenous allele of ngSeipin. Thus, *ELife* set as a condition for further consideration of the manuscript that the authors use modern tools of gene editing to re-create the disease-associated ngSEIPIN heterozygous state in a suitable cellular system (of their choice) and confirm that the phenotype arising from this physiologically expressed model approaches the consequences of the over-expression system used here in terms of calcium handling, association of ngSEIPIN with SERCA and its oligomerisation.

2) Much rests on the claim that ngSeipin is more aggregation prone than the wild type. Yet the experimental evidence for this rests on interpretation of a proximity based ligation assay, which is deemed by our expert reviewers to be a weak assay for this purpose. Therefore, this key conclusion must be convincingly buttressed by an assay orthogonal to PLA.

3) The consensus amongst the reviewers is that the paper has not shown that MND caused by ngSeipin is a consequence of the calcium perturbation and ER stress. Thus, the title, which suggests this causative relationship should be revised. A more apt title would make the case for a correlation, for example: "The MND-associate mutation in Seipin encodes a protein that inhibits SERCA, etc."

*Reviewer #1 (Recommendations for the authors):*

To improve the manuscript, and its suitability for *eLife*, two aspects should be included/discussed:

1. The authors should (at least) discuss why in wtSeipin deletion of the N-terminus and modification of the TM helices has a very large effect on ER calcium levels (Figure 6C). A mechanistic explanation would be a plus.

2. As a complement to Figure 3-S1D(c), the authors should also provide the number of LD per cell, in addition to their diameter. In fact, seipin-KO cells normally result in a phenotype with few supersized LDs and many very small LD. However, quite surprisingly, in Figure3-S1D(c) no such supersized LDs are observed even though they can be spotted in Figure3-S1D(a).

*Reviewer #2 (Recommendations for the authors):*

1. Seipin has two transmembrane segments with an extended ER luminal domain and the N and C termini in the cytosol. Defining it as a "hairpin-like transmembrane protein in the ER" is misleading and likely incorrect.

2. Figure 1D. The difference in the amounts of wt and ngSeipin co-precipitated by SERCA are unimpressive, considering the overexpression conditions mentioned above and the lack of a specificity control.

3. The calcium changes are modest. It would be interesting to see whether endogenous SERCA localization changes upon expression of wt and ngSeipin.

4. The expression levels of some proteins are not well described. For example, in figure 4C all Seipin mutants appear to be expressed to the same level (even if a loading control is not shown) however in Figure 5A ngSeipin M6 looks much less abundant.

5. Figure 3Ab, shows that the best rescue of Seipin KO cells with WT seipin is observed with low DNA transfections (0.52ng). It is unclear why much larger amounts (10.4ng) are used in most experiments.

6. The rescue of calcium levels observed by SERCA overexpression is modest and there are alternatives for the experiment. For example, WT and mutant SERCA can be expressed to different levels, as these have not been examined.

*Reviewer #3 (Recommendations for the authors):*

1. Seipinopathy is a rare neurodegenerative disease. An introduction or discussion about current evidence linking cellular calcium homeostasis with neurodegenerative diseases may help to further strengthen the manuscript and draw broader interest.

2. The authors provided extensive biochemical and genetic evidence from HCT116 cells for the function of non-glycosylated Seipin in ER calcium homeostasis, ER stress, cell apoptosis. Seipinopathy-causal mutants can be directly created by CRISPR/Cas9 system. Further studies in neuron cells or SH-SY5Y cells will help to understand the role of non-glycosylated Seipin in Seipinopathy

3. An important finding for this manuscript is that non-glycosylated Seipin causes ER stress and cell apoptosis through inactivating ER calcium pump SERCA. A more detailed method for how ER or cytosol calcium levels was quantified should be provided. The name of ER ca^2+^ indicator is "G-CEPIA1er" but not "Cepia1er". The authors should clarify it and keep it consistent in the manuscript. Literature associated with ca^2+^ quantification and ca^2+^ indicators applied should also be cited in the manuscript.

4. A short exposure image for Myc-wt/ngSeipin blots in Figure1E(c) may help to interpret the result.

5. The authors showed that overexpression of SERCA2b, but not SERCA2b(Q108H), restored calcium levels in ER. SERCA2b(Q108H) expression is also suggested to be set as a negative control in the following cleaved caspase 3 assays of Figure 9D.

6. The black and red bars in Figure 10 should be annotated in the main Figure instead of in the figure legends.

[Editors' note: further revisions were suggested prior to acceptance, as described below.]

Thanks for submitting a revised version of your paper 'A Motor Neuron Disease-associated Mutation Produces Non-glycosylated Seipin that Induces ER Stress and Apoptosis by Inactivating SERCA2b'.

I have looked over your revised version and compared it to the decision letter and am satisfied that in all respects but one your paper addresses the key points. The remaining point of uncertainty relates to the extent to which the differences observed between the wildtype and disease-associated mutant Seipin are manifest in a physiologically relevant concentration regime. This issue was flagged in the review as the most critical. To recapitulate: the reviewers accepted your claim that expression of the wildtype and mutant proteins have different consequences in terms of ER calcium metabolism and stress response. They also accepted that these may be plausibly related to different biophysical properties of the two proteins. However, for these features to be plausibly linked to the disease phenotype, they must be manifest at a physiological concentration regime. In other words, the reviewers were legitimately concerned by the possibility that the very real differences you observe between the wildtype and mutant were dependent on an abnormally high concentration in the cell and thus represent an over-expression experimental artifact of uncertain relevance.

The reviewers insisted that this issue must be addressed by studying the consequences of the mutation at endogenous levels of expression and suggested modifying the endogenous Seipin gene to that end. From your cover letter, I understand that this was attempted, but failed. The alternative approach you took – to cover the null with wild-type or mutant Seipin, in trans – is also acceptable, in principle. However, it remains essential to be able to relate the levels of expression achieved by this alternative approach to the endogenous levels of expression in the cell line in which the experiments were performed (SH SY5Y cells in this case). Perhaps I missed it, but to my reading Figure 10 shows comparability in expression of the wildtype and mutant Seipin transgenic protein in the Seipin∆ cells. I could find no assay that compares this to the expression of endogenous Seipin in the parental wild-type cells.

If I missed this critical piece of data, kindly direct me to it.

If the measurement is missing from the paper, please endeavour to complete it and submit a revised version with this critical piece.

---

## [Author Response]

Essential revisions:1) The reviewers were divided in their perception of the significance attributed to the changes in SERCA activity and calcium metabolism observed in cells expressing wild type and ngSeipin. However, in consultation a consensus emerged whereby it was deemed essential that the perturbations arising from ectopic expression of ngSeipin were also examined in a system mimicking the genetics of the diseases namely a mutant endogenous allele of ngSeipin. Thus, ELife set as a condition for further consideration of the manuscript that the authors use modern tools of gene editing to re-create the disease-associated ngSEIPIN heterozygous state in a suitable cellular system (of their choice) and confirm that the phenotype arising from this physiologically expressed model approaches the consequences of the over-expression system used here in terms of calcium handling, association of ngSEIPIN with SERCA and its oligomerisation.

We intended to knock-in N152S or S154L mutation in SH-SY5Y cells using

ABE (doi: 10.1038/nature24644.),

BE4-Gam (doi: 10.1126/sciadv.aao4774),

CRISPR/Cas9 + ssODN (doi: 10.1038/nbt.3481.),

MhAX method (doi: 10.1038/s41467-018-03044-y)

but failed to do so, possibly due to its slow growth rate (doubling time = ~67 h) (Feles et al., 2022) and low transfection efficiency (10-15%).

Instead, we obtained Seipin-KO SH-SY5Y cells using the CRISPR/Cas9 technology and showed that ngSeipin expressed by plasmid transfection at an endogenous protein level, as estimated by quantification of immunoblotting results and correction of transfection efficiency, decreased [ca^2+^] in the ER, distorted morphology of the ER (consequence of aggregation), induced ER stress, and induced apoptosis. Furthermore, simultaneous expression of ngSeipin and wtSeipin by transfection each at a half amount of the above plasmid evoked the same phenotype (text p. 19-21, Figure 10).

We will keep working on knock-in experiments and hope to publish the positive results as a Research Advance in the near future.

2) Much rests on the claim that ngSeipin is more aggregation prone than the wild type. Yet the experimental evidence for this rests on interpretation of a proximity based ligation assay, which is deemed by our expert reviewers to be a weak assay for this purpose. Therefore, this key conclusion must be convincingly buttressed by an assay orthogonal to PLA.

We have carried out immunofluorescence analysis and shown that expression of ngSeipin but not wtSeipin distorted morphology of the ER and altered the localization of SERCA2b, producing double (Seipin & SERCA2b)-negative regions in the ER of enlarged cells. Expression of ngSeipin(DC) but not ngSeipin(M6) produced double-negative regions in the ER. These results are consistent with those of PLA. Thus, ngSeipin is aggregation-prone and oligomerization-dependent aggregation of ngSeipin incorporates SERCA2b into aggregates, resulting in inactivation of SERCA2b (text p. 16-18, Figure 8, 8-S1, 8-S2).

3) The consensus amongst the reviewers is that the paper has not shown that MND caused by ngSeipin is a consequence of the calcium perturbation and ER stress. Thus, the title, which suggests this causative relationship should be revised. A more apt title would make the case for a correlation, for example: "The MND-associate mutation in Seipin encodes a protein that inhibits SERCA, etc."

The title has been changed to “A motor neuron disease-associated mutation produces non-glycosylated Seipin that induces ER stress and apoptosis by inactivating SERCA2b”

Reviewer #1 (Recommendations for the authors):To improve the manuscript, and its suitability for eLife, two aspects should be included/discussed:1. The authors should (at least) discuss why in wtSeipin deletion of the N-terminus and modification of the TM helices has a very large effect on ER calcium levels (Figure 6C). A mechanistic explanation would be a plus.

We have shown that the distorted morphology of the ER explained the unexpected decrease in [ca^2+^] in the ER in Seipin-KO cells expressing Seipin^L^(DN), Seipin^L^(TM1), and Seipin^L^(TM2) by transfection (Figure 6D, bars 5, 7, and 10). Such cells contained double-negative regions similarly to Seipin-KO cells expressing ngSeipin^L^ (Figure 8-S1A(a)(b), compare bars 4, 6, and 8 with bar 2), and the percentage of cells with double-negative regions increased in Seipin-KO cells expressing their respective non-glycosylated version (Figure 8-S1A(a)(b), bars 5, 7. and 9). It is likely that truncation of the N-terminal region or swapping of the transmembrane domain per se adversely affected the structural maintenance of Seipin, making the Seipin mutants aggregation-prone.

2. As a complement to Figure 3-S1D(c), the authors should also provide the number of LD per cell, in addition to their diameter. In fact, seipin-KO cells normally result in a phenotype with few supersized LDs and many very small LD. However, quite surprisingly, in Figure3-S1D(c) no such supersized LDs are observed even though they can be spotted in Figure3-S1D(a).

We have deleted the data on lipid droplets from the revised manuscript and will report them elsewhere as a separate paper.

Reviewer #2 (Recommendations for the authors):1. Seipin has two transmembrane segments with an extended ER luminal domain and the N and C termini in the cytosol. Defining it as a "hairpin-like transmembrane protein in the ER" is misleading and likely incorrect.

We have changed "hairpin-like transmembrane protein in the ER" to “a protein which spans the ER membrane twice”.

2. Figure 1D. The difference in the amounts of wt and ngSeipin co-precipitated by SERCA are unimpressive, considering the overexpression conditions mentioned above and the lack of a specificity control.

This would be due to incomplete solubilization of aggregation-prone ngSeipin in 1% NP-40.

3. The calcium changes are modest. It would be interesting to see whether endogenous SERCA localization changes upon expression of wt and ngSeipin.

Please see our response to Essential revisions 2.

4. The expression levels of some proteins are not well described. For example, in figure 4C all Seipin mutants appear to be expressed to the same level (even if a loading control is not shown) however in Figure 5A ngSeipin M6 looks much less abundant.

We think it is within the range of experimental variation.

5. Figure 3Ab, shows that the best rescue of Seipin KO cells with WT seipin is observed with low DNA transfections (0.52ng). It is unclear why much larger amounts (10.4ng) are used in most experiments.

Transfection at > 2.60 ng/cm^2^ is necessary for ngSeipin to decrease [ca^2+^] in the ER (Figure 3A). We showed the dose-dependence of ngSeipin expression in most experiments (Figure 3, Figure 4, Figure 8). Transfection at 10.4 ng/cm^2^ was used to show the negative effects of ngSeipin(M6) (Figure 5), Seipin (DLD) and Seipin(Cterm) (Figure 6), and ngSeipin(DC) (Figure 7) as well as the positive effect of SERCA2b overexpression (Figure 9).

6. The rescue of calcium levels observed by SERCA overexpression is modest and there are alternatives for the experiment. For example, WT and mutant SERCA can be expressed to different levels, as these have not been examined.

Such overexpression sufficiently decreased ngSeipin-mediated induction of ER stress and apoptosis (Figure 9).

Reviewer #3 (Recommendations for the authors):1. Seipinopathy is a rare neurodegenerative disease. An introduction or discussion about current evidence linking cellular calcium homeostasis with neurodegenerative diseases may help to further strengthen the manuscript and draw broader interest.

We will do so in a future paper.

2. The authors provided extensive biochemical and genetic evidence from HCT116 cells for the function of non-glycosylated Seipin in ER calcium homeostasis, ER stress, cell apoptosis. Seipinopathy-causal mutants can be directly created by CRISPR/Cas9 system. Further studies in neuron cells or SH-SY5Y cells will help to understand the role of non-glycosylated Seipin in Seipinopathy

Please see our response to Essential revisions 1.

3. An important finding for this manuscript is that non-glycosylated Seipin causes ER stress and cell apoptosis through inactivating ER calcium pump SERCA. A more detailed method for how ER or cytosol calcium levels was quantified should be provided.

Such a method is described on p. 30.

The name of ER ca^2+^ indicator is "G-CEPIA1er" but not "Cepia1er". The authors should clarify it and keep it consistent in the manuscript.

"Cepia1er" has been changed to "G-CEPIA1er".

Literature associated with ca^2+^ quantification and ca^2+^ indicators applied should also be cited in the manuscript.

Such references are cited in p. 8, line 9.

4. A short exposure image for Myc-wt/ngSeipin blots in Figure1E(c) may help to interpret the result.

We have done so.

5. The authors showed that overexpression of SERCA2b, but not SERCA2b(Q108H), restored calcium levels in ER. SERCA2b(Q108H) expression is also suggested to be set as a negative control in the following cleaved caspase 3 assays of Figure 9D.

6. The black and red bars in Figure 10 should be annotated in the main Figure instead of in the figure legends.

We have done so.

[Editors' note: further revisions were suggested prior to acceptance, as described below.]

I have looked over your revised version and compared it to the decision letter and am satisfied that in all respects but one your paper addresses the key points. The remaining point of uncertainty relates to the extent to which the differences observed between the wildtype and disease-associated mutant Seipin are manifest in a physiologically relevant concentration regime. This issue was flagged in the review as the most critical. To recapitulate: the reviewers accepted your claim that expression of the wildtype and mutant proteins have different consequences in terms of ER calcium metabolism and stress response. They also accepted that these may be plausibly related to different biophysical properties of the two proteins. However, for these features to be plausibly linked to the disease phenotype, they must be manifest at a physiological concentration regime. In other words, the reviewers were legitimately concerned by the possibility that the very real differences you observe between the wildtype and mutant were dependent on an abnormally high concentration in the cell and thus represent an over-expression experimental artifact of uncertain relevance.The reviewers insisted that this issue must be addressed by studying the consequences of the mutation at endogenous levels of expression and suggested modifying the endogenous Seipin gene to that end. From your cover letter, I understand that this was attempted, but failed. The alternative approach you took – to cover the null with wild-type or mutant Seipin, in trans – is also acceptable, in principle. However, it remains essential to be able to relate the levels of expression achieved by this alternative approach to the endogenous levels of expression in the cell line in which the experiments were performed (SH SY5Y cells in this case). Perhaps I missed it, but to my reading Figure 10 shows comparability in expression of the wildtype and mutant Seipin transgenic protein in the Seipin∆ cells. I could find no assay that compares this to the expression of endogenous Seipin in the parental wild-type cells. If I missed this critical piece of data, kindly direct me to it.If the measurement is missing from the paper, please endeavour to complete it and submit a revised version with this critical piece. For my part, I'll accelerate the handling of the paper as much as I can, as I realise it has now been under review for over one year.

We have added quantified data of immunoblotting to Figure 10A(c).

We have estimated transfection efficiency to be approximately 15% by transfecting

SH-SY5Y cells with plasmid (104 ng/cm^2^) to express G-CEPIA1er. We have included these data in Figure 10-source data 1.

Accordingly, we have rewritten the text as follows (p. 19, line 1 from bottom – p. 20, line 4).

These expression levels were estimated to be approximately 16% of those of endogenous Seipin^L^ and Seipin^S^ [Figure 10A(c)]. Given a transfection efficiency of approximately 15%, as estimated by transfecting SH-SY5Y cells with plasmid (104 ng/cm^2^) to express G-CEPIA1er (Figure 10-source data 1), the levels of ngSeipin^L^ and ngSeipin^S^ expressed by transfection in Seipin-KO SH-SY5Y cells would be considered comparable to those of endogenous Seipin^L^ and Seipin^S^ in SH-SY5Y WT cells.